# HYPERBOLIC GENOME EMBEDDINGS

**Raiyan R. Khan, Philippe Chlenski, Itsik Pe'er**
Columbia University
`{raiyan, pac, itsik}@cs.columbia.edu`

## ABSTRACT

Current approaches to genomic sequence modeling often struggle to align the inductive biases of machine learning models with the evolutionarily-informed structure of biological systems. To this end, we formulate a novel application of hyperbolic CNNs that exploits this structure, enabling more expressive DNA sequence representations. Our strategy circumvents the need for explicit phylogenetic mapping while discerning key properties of sequences pertaining to core functional and regulatory behavior. Across 37 out of 42 genome interpretation benchmark datasets, our hyperbolic models outperform their Euclidean equivalents. Notably, our approach even surpasses state-of-the-art performance on seven GUE benchmark datasets, consistently outperforming many DNA language models while using orders of magnitude fewer parameters and avoiding pretraining. Our results include a novel set of benchmark datasets—the Transposable Elements Benchmark—which explores a major but understudied component of the genome with deep evolutionary significance. We further motivate our work by exploring how our hyperbolic models recognize genomic signal under various data-generating conditions and by constructing an empirical method for interpreting the hyperbolicity of dataset embeddings. Throughout these assessments, we find persistent evidence highlighting the potential of our hyperbolic framework as a robust paradigm for genome representation learning. Our code and benchmark datasets are available at `https://github.com/rrkhan/HGE`.

## 1 INTRODUCTION

Representation learning of genome sequences has enabled the exploration of critical unsolved problems in biology, particularly the understanding of genome function and organization (Avsec et al., 2021; Chen et al., 2022a; Dudnyk et al., 2024). Many effective approaches used for genomic sequence modeling have arisen from the same machine learning methods that have powered natural language and image embeddings (Yue et al., 2023; Zhou, 2022; Consens et al., 2023). While the field has made progress by utilizing these methods, the inductive biases of these models are not usually bespoke to genomic data, limiting the expressive power of the resulting sequence representations. Given the tremendous amount of information sequestered within DNA sequences encoding cellular and molecular activity, an efficient and nuanced representation is necessary for genome interpretation and downstream analyses.

Genome organization is complex, and much of this complexity is the product of evolutionary processes. Any single genome represents the culmination of information diffusion across generations. However, this information transfer occurs through noisy channels, as background mutation rates may degrade the sequence signal (Lu et al., 2020). Accounting for phylogenetic relationships may therefore contextualize the content of the genome and ultimately benefit genome interpretation attempts. The shared influence of a common ancestor across all genomes imbues DNA sequence data with underlying hierarchical structure. These hierarchical relationships emerge through a variety of mechanisms, such as orthology and paralogy, which both codify homologous sequences but occur under different circumstances. Further compounding these interdependencies are the multiple overlapping grammatical structures for different regulatory pathways that characterize the language of the genome. Altogether, these nested levels of latent hierarchies confound genome interpretation.

In developing a modeling paradigm better suited to handling the hierarchical nature of DNA sequences, considering the geometry of the embedding spaces is essential. While most embeddings are Euclidean

by default, non-Euclidean spaces may offer a compelling alternative. Specifically, hyperbolic spaces, which have the representational capacity to capture tree-structured data with high fidelity, are well-equipped to manage the hierarchical patterns ubiquitous in genomic sequences. The negative curvature of hyperbolic spaces facilitates the continuous embedding of exponentially growing structures like phylogenetic trees with relatively low distortion.

In this work, we contend that hyperbolic spaces may be appropriate for learning meaningful representations of the genome. We leverage a fully hyperbolic framework to embed DNA sequences, implicitly handling the latent hierarchies present in the data. Our main contributions are:

1. We adopt the machinery of fully hyperbolic convolutional neural networks (HCNNs), building two classes of HCNNs for genome sequence learning. We contrast hyperbolic and Euclidean approaches to sequence representation.
2. We introduce a novel, curated set of datasets—the Transposable Elements Benchmark—designed to investigate transposable elements, which remain an underexplored area of the genome with deep evolutionary roots.
3. We demonstrate the performance potential of our HCNNs across 42 real-world datasets addressing foundational challenges in genomics.
4. We elucidate the underlying mechanism by which HCNNs parse genomic signal by simulating and testing plausible data-generating processes for biological sequences.
5. We further motivate our work by formulating an empirical method for interpreting the hyperbolicity of dataset embeddings and use this technique to interrogate properties of genome representations generated by our models.

## 2 PRELIMINARIES

### 2.1 RELATED WORK

Driven by the limitations of traditional Euclidean-based approaches in capturing relationships within complex data structures, hyperbolic deep learning methods have materialized as a promising research area. Early iterations of these methods introduced formalizations for performing the core operations of neural networks in hyperbolic space (Ganea et al., 2018; Nickel & Kiela, 2018), alongside optimization techniques generalized to Riemannian manifolds (Bécigneul & Ganea, 2019). These approaches have been further extended to a variety of frameworks, including fully hyperbolic neural networks (Chen et al., 2022b), hyperbolic graph convolutional networks (Chami et al., 2019), hyperbolic attention networks (Gulcehre et al., 2018), and hyperbolic variational auto-encoders (Mathieu et al., 2019). These models, among others, have proven effective across a variety of real-world domains, including vision (Liu et al., 2020; Hsu et al., 2021; Mathieu et al., 2019), natural language (Tifrea et al., 2019; Chen et al., 2024), and computational biology (Zhou & Sharpee, 2021; Tian et al., 2023).

In genomics, hyperbolic methods have correctly modeled established phylogenies, showcasing their supremacy in representing tree-structured data (Chami et al., 2020a; Jiang et al., 2022b; Hughes et al., 2004; Chen et al., 2025). These methods assume that the phylogenetic tree is known *a priori*; thus, the scope of these techniques is limited by the availability of evolutionary metadata. A subset of these methods produces representations of DNA sequences but relies on an explicit mapping of phylogenetic relationships (Corso et al., 2021; Jiang et al., 2022a) in the form of pairwise edit distances or incomplete phylogenies.

### 2.2 BACKGROUND

The $n$-dimensional hyperbolic space $\mathbb{H}_K^n$ is a homogeneous, simply connected Riemannian manifold, $(\mathcal{M}^n, \mathfrak{g}_x^K)$, consisting of a smooth manifold $\mathcal{M}^n$ with Riemannian metric $\mathfrak{g}_x^K$, and described by a constant negative curvature $K < 0$. Several equivalent formulations of hyperbolic space exist, including the Lorentz model, the Poincaré disk model, and the (Beltrami-)Klein model. Here, we use the Lorentz model, $\mathbb{L}_K^n = (\mathcal{L}^n, \mathfrak{g}_x^K)$, with manifold $\mathcal{L}^n$, Riemannian metric tensor $\mathfrak{g}_x^K = \mathrm{diag}(-1, 1, \ldots, 1)$, and origin $\overline{\mathbf{0}} = [\sqrt{-1/K}, 0, \ldots, 0]^T$. The Lorentz model describes points by their configurations on the forward sheet of a two-sheeted hyperboloid $\mathbb{L}_K^n$ in $(n+1)$-dimensional

Minkowski space, defining the manifold as:

$$\mathcal{L}^n := \{\boldsymbol{x} \in \mathbb{R}^{n+1} \mid \langle \boldsymbol{x}, \boldsymbol{x} \rangle_{\mathcal{L}} = \frac{1}{K}, x_t > 0\}, \tag{1}$$

where the Lorentzian inner product is as follows:

$$\langle \boldsymbol{x}, \boldsymbol{y} \rangle_{\mathcal{L}} := -x_t y_t + \boldsymbol{x}_s^T \boldsymbol{y}_s = \boldsymbol{x}^T \text{diag}(-1, 1, \ldots, 1) \boldsymbol{y}. \tag{2}$$

Utilizing special relativity conventions, the zeroth element in $\boldsymbol{x}$ is denoted as the timelike component $x_t$ and the remaining $n-1$ elements form the spacelike components $\boldsymbol{x}_s$, giving $\boldsymbol{x} = [x_t, \boldsymbol{x}_s]^T$, where we can further define the timelike component $x_t = \sqrt{||\boldsymbol{x}_s||^2 - 1/K}$.

Exponential and logarithmic maps are used to map between the manifold $\mathcal{M}$ and tangent space $T_{\boldsymbol{x}}\mathcal{M}$ with $\boldsymbol{x} \in \mathcal{M}$. For mapping a tangent vector $\boldsymbol{z} \in T_{\boldsymbol{x}}\mathbb{L}_K^n$ onto the Lorentz manifold, we can use the exponential map which is defined as:

$$\exp_{\boldsymbol{x}}^K(\boldsymbol{z}) = \cosh(\alpha)\boldsymbol{x} + \sinh(\alpha)\frac{\boldsymbol{z}}{\alpha}, \quad \text{where } \alpha = \sqrt{-K}||\boldsymbol{z}||_{\mathcal{L}}, \quad ||\boldsymbol{z}||_{\mathcal{L}} = \sqrt{\langle \boldsymbol{z}, \boldsymbol{z} \rangle_{\mathcal{L}}}. \tag{3}$$

Conversely, to map a point $\boldsymbol{y} \in \mathbb{L}_K^n$ to the tangent space, we use the logarithmic map:

$$\log_{\boldsymbol{x}}^K(\boldsymbol{y}) = \frac{\cosh^{-1}(\beta)}{\sqrt{\beta^2 - 1}} \cdot (\boldsymbol{y} - \beta\boldsymbol{x}), \quad \beta = K\langle \boldsymbol{x}, \boldsymbol{y} \rangle_{\mathcal{L}}. \tag{4}$$

Furthermore, in order to move points along geodesics, the parallel transport operation $\text{PT}_{\boldsymbol{x} \to \boldsymbol{y}}^K(\boldsymbol{v})$ maps a vector $\boldsymbol{v} \in T_{\boldsymbol{x}}\mathcal{M}$ from the tangent space of $\boldsymbol{x} \in \mathcal{M}$ to the tangent space of $\boldsymbol{y} \in \mathcal{M}$. The Lorentzian formula for parallel transport is:

$$\text{PT}_{\boldsymbol{x} \to \boldsymbol{y}}^K(\boldsymbol{v}) = \boldsymbol{v} + \frac{\langle \boldsymbol{y}, \boldsymbol{v} \rangle_{\mathcal{L}}}{\frac{1}{-K} - \langle \boldsymbol{x}, \boldsymbol{y} \rangle_{\mathcal{L}}}(\boldsymbol{x} + \boldsymbol{y}). \tag{5}$$

## 3 METHODS

### 3.1 FULLY HYPERBOLIC CNN

We leverage the HCNN methodology proposed by Bdeir et al. (2024) in the development of our fully hyperbolic genome sequence model. Under this framework, the elements of the traditional CNN model are reinterpreted in the context of the Lorentz model of hyperbolic space. Briefly, we describe the main Lorentzian components utilized in our model.

**Lorentz Convolutional Layer.** In a Euclidean setting, a convolutional layer constitutes matrix multiplication between a linearized kernel and input feature maps. In the hyperbolic analogue, each channel is defined as a separate point on the hyperboloid, with the input to each layer forming an ordered set of $n$-dimensional hyperbolic vectors in $\mathbb{L}_K^n$. This formulation enforces the constraint that operations on points remain on the hyperboloid, as $\mathbb{L}_K^n \subset \mathbb{R}^{n+1}$. In the context of this work, each sequence is thus an ordered set of $n$-dimensional hyperbolic vectors, where each position describes a nucleotide in the sequence.

For a one-dimensional hyperbolic convolutional layer with input feature map $\boldsymbol{x} = \{\boldsymbol{x}_l \in \mathbb{L}_K^n\}_{l=1}^L$, the features contained in the receptive field of kernel $\mathbf{G} \in \mathbb{R}^{m \times n \times \tilde{L}}$ are $\{\boldsymbol{x}_{l' + \epsilon\tilde{l}} \in \mathbb{L}_K^n\}_{\tilde{l}=1}^{\tilde{L}}$, in which $l'$ marks the starting position and $\epsilon$ is the stride. Given this parameterization, we can express the convolution layer as the output of two transformations:

$$\boldsymbol{y}_l = \text{LFC}(\text{HCat}(\{\boldsymbol{x}_{l' + \epsilon\tilde{l}} \in \mathbb{L}_K^n\}_{\tilde{l}=1}^{\tilde{L}})), \tag{6}$$

where HCat is an operation concatenating hyperbolic vectors, and LFC is a Lorentz fully-connected layer performing the affine transformation of the kernel (refer to A.1.1). Next, **Lorentz batch normalization** (LBN) reframes the underlying operations of batch normalization by using Fréchet mean (Lou et al., 2020) for re-centering points and Fréchet variance (Kobler et al., 2022) for re-scaling them. The algorithm is expressed as:

$$\text{LBN}(\boldsymbol{x}) = \exp_{\boldsymbol{\beta}}^K \left( \text{PT}_{\mathbf{0} \to \boldsymbol{\beta}}^K \left( \gamma \cdot \frac{\text{PT}_{\boldsymbol{\mu}_B \to \overline{\mathbf{0}}}^K \left( \log_{\boldsymbol{\mu}_B}^K(\boldsymbol{x}) \right)}{\sqrt{\sigma_B^2 + \epsilon}} \right) \right). \tag{7}$$

Finally, **Lorentz multinomial logistic regression (MLR)** builds upon the original formulation of a Euclidean MLR (Lebanon & Lafferty, 2004), which is defined using input $\boldsymbol{x} \in \mathbb{R}^n$ and $C$ classes:

$$p(y = c|\boldsymbol{x}) \propto \exp(v_{\boldsymbol{w}_c}(\boldsymbol{x})), \quad v_{\boldsymbol{w}_c}(\boldsymbol{x}) = \text{sign}(\langle \boldsymbol{w}_c, \boldsymbol{x} \rangle) \|\boldsymbol{w}_c\| d(\boldsymbol{x}, H_{\boldsymbol{w}_c}), \quad \boldsymbol{w}_c \in \mathbb{R}^n, \quad (8)$$

in which $H_{\boldsymbol{w}_c}$ is the decision hyperplane of class $c$. Bdeir et al. (2024) replace component operations with their Lorentzian interpretations to produce the Lorentz MLR formulation. Using parameters $a_c \in \mathbb{R}$ and $\boldsymbol{z}_c \in \mathbb{R}^n$, the Lorentz MLR's output logit for class $c$ given input $\boldsymbol{x} \in \mathbb{L}_K^n$ is expressed as:

$$v_{\boldsymbol{z}_c, a_c}(\boldsymbol{x}) = \frac{1}{\sqrt{-K}} \text{sign}(\alpha) \beta \left| \sinh^{-1} \left( \sqrt{-K} \frac{\alpha}{\beta} \right) \right|, \quad (9)$$

$$\alpha = \cosh(\sqrt{-K}a) \langle \boldsymbol{z}, \boldsymbol{x}_s \rangle - \sinh(\sqrt{-K}a),$$

$$\beta = \sqrt{\| \cosh(\sqrt{-K}a) \boldsymbol{z} \|^2 - (\sinh(\sqrt{-K}a) \|\boldsymbol{z}\|)^2}.$$

For further details, including Lorentz formulations of residual connections and nonlinear activation, we refer the reader to Bdeir et al. (2024).

## 3.2 MODEL OVERVIEW

As our goal is to distill the difference between using Euclidean versus hyperbolic embedding spaces, we employ a relatively simple model design. The HCNN architecture consists of three major components: (1) hyperbolic convolutional blocks, (2) a flattening layer, and (3) MLR (Figure 1). Each input DNA sequence $\mathbf{x}$ is one-hot encoded at the nucleotide level, and then projected channel-wise onto a hyperbolic manifold ($\varphi : \mathbb{R}^{4 \times L} \to \mathbb{L}^{4 \times L}$). The result of this transformation serves as the input to the hyperbolic convolutional blocks, which produce output feature maps $\mathbf{x} \in \mathbb{L}^{C \times L}$, where $C$ is the channel dimension. After a flattening step, the model performs classification using Lorentz MLR to find the hyperbolic decision hyperplanes splitting the sequences by label.

For each hyperbolic component in our models, there exists an equivalent Euclidean counterpart, ensuring architectural parity across models for a fair comparison (Appendix Figure 4). However, the layers in the HCNNs also include a learnable $K$ parameter corresponding to the curvature of the hyperboloid on which the points reside. For our downstream experiments, we evaluate two versions of the HCNN model: HCNN-S (single $K$) and HCNN-M (multiple $K$s). In HCNN-S, a single manifold with a fixed curvature $K$ is used across all layers, offering a more direct comparison with CNNs. In contrast, HCNN-M assigns distinct curvatures $[K_1, ..., K_u]$ to each of the $u$ designated blocks, with intermediary steps mapping points between manifolds (see A.1.2). By constructing two classes of HCNN models, we analyze the trade-offs between the enhanced representational flexibility of multiple curvatures and the potential instability introduced by the additional exponential and logarithmic mapping steps required for projecting points onto different manifolds. Additional modeling details are provided in A.2.

## 3.3 $\delta$-HYPERBOLICITY

Gromov introduces the notion of $\delta$-hyperbolicity as a measure of the deviation of a metric space from perfect tree-like structure (Gromov, 1987). We can define a metric space $(M, d)$, in which the Gromov product of $z, y \in M$ with respect to $x \in M$ is:

$$(x, y)_z = \frac{1}{2} \left( d(x, z) + d(y, z) - d(x, y) \right). \quad (10)$$

Then, the metric space is characterized as $\delta$-hyperbolic for some $\delta \geq 0$ if it satisfies the *four point condition*, which states that for any four points $x, y, z, w \in M$:

$$(x, y)_w \geq \min\{(x, z)_w, (y, z)_w\} - \delta. \quad (11)$$

The smallest $\delta$ for which this inequality holds is the Gromov $\delta$-hyperbolicity of $(M, d)$.

$\delta$-hyperbolicity has been an important tool in elucidating innate properties of metric spaces (Fournier et al., 2015; Albert et al., 2014). Recently, this measure has been extended to explore the hyperbolic

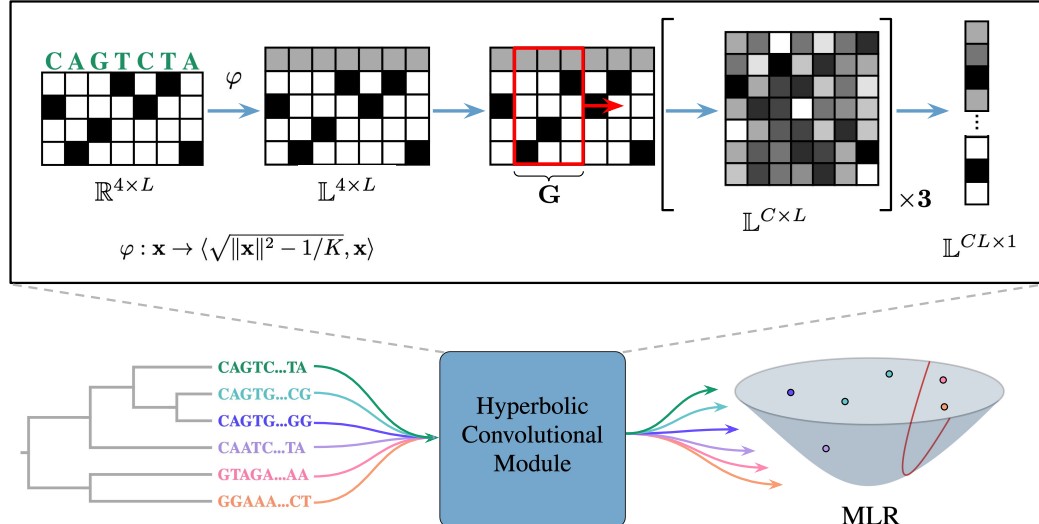

Figure 1: Overview of our HCNNs. Model inputs are sequences with latent phylogenetic structure (bottom left). As sequences pass through the hyperbolic convolutional module, they are projected onto a hyperboloid before the model's convolutional and flattening steps (top insert). Using hyperbolic MLR, each sequence is classified according to the hyperplane boundaries (bottom right).

behavior of specific datasets and their respective embeddings within the domains of computer vision (Khrulkov et al., 2020), and natural language processing (Yang et al., 2024). While the original Gromov's $\delta$ (which we will denote as $\delta_{\text{worst}}$ hereinafter) is designed to represent the upper bound in terms of deviation from tree-like structure, other approaches have argued in favor of utilizing an average Gromov hyperbolicity, $\delta_{\text{avg}}$, on the grounds that a worst case analysis of a space may not ultimately be representative of the true hyperbolic capacity of the space (Chatterjee & Sloman, 2021; Albert et al., 2014; Tifrea et al., 2019). We further develop these ideas in the context of the genomic datasets used in this paper.

As in previous approaches, we examine the behavior of $\delta_{\text{worst}}$ and $\delta_{\text{avg}}$ in high dimensional feature space. As a comparative measure, we compute a scale-invariant value of $\delta$, defined as $\delta_{\text{rel}} := \frac{2\delta}{D_{\text{max}}}$ (Borassi et al., 2015), where $D_{\text{max}}$ denotes the maximal pairwise distance, or set diameter. $\delta_{\text{rel}}$ is constrained to $[0, 1]$, with a value of 0 denoting complete hyperbolicity, or perfect tree structure. Unless otherwise specified, all $\delta$s referred to in this work are the scale-invariant value.

Ultimately, both $\delta_{\text{worst}}$ and $\delta_{\text{avg}}$ are point estimates over what may be a complex landscape of $\delta$ values. To offer a more comprehensive evaluation, we examine the entire distribution of $\delta$ values across each dataset to thoroughly assess the hyperbolic underpinnings of DNA sequence data. By appraising the full landscape of $\delta$-hyperbolicity in our embedding space, we gain a richer understanding of the intrinsic tree structure across each dataset. We provide further details on $\delta$ computation and other experimental configurations in A.9.1.

## 4 DATA

**Synthetic Datasets.** In order to rigorously interrogate the applicability of hyperbolic architectures in genomics, we create several synthetic datasets to illuminate the underlying biological processes being captured by our models. Our approach considers various plausible data-generating processes for biological sequences, and defines three potential cases of biological signal transmission learned by the models. Additionally, given prior evidence from Corso et al. (2021) that purely artificial sequences may not always be indicative of performance on real-world datasets, we explore this phenomenon by creating two sets of data for each case: one where sequences are completely randomly generated and one where sequences are randomly sampled from existing genomes.

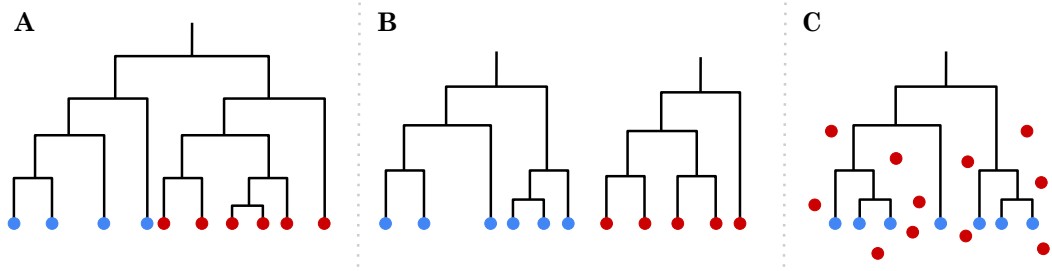

Figure 2: The various plausible evolutionary scenarios informing genomic sequence learning. Leaf coloring (blue vs. red) represents label assignments for A) intra-tree differentiation, B) inter-tree differentiation, and C) tree identification scenarios.

We mimic evolutionary dynamics in our synthetic datasets by perturbing input sequences based on phylogenetic tree structure. After establishing an initial input sequence, we simulate sequence evolution along tree branches with the generalized time-reversible (GTR) nucleotide model (Tavaré, 1984). Each scenario, visualized in Figure 2, is defined as follows:

(A) **Intra-tree differentiation**: sequences are generated from a single phylogenetic tree, with labels assigned based on clade membership.
(B) **Inter-tree differentiation**: sequences are generated from different phylogenetic trees, with labels derived from phylogeny membership.
(C) **Tree identification**: sequences are labeled based on the generating process: phylogenetic tree generation or non-phylogenetic (random) generation.

We leverage these scenarios to better understand the specific advantages of hyperbolic models and identify the conditions under which they demonstrate the greatest effectiveness. For full details regarding dataset generation, see A.6.

**Transposable Elements Benchmark.** We introduce a multi-species benchmark for exploring how transposable elements (TEs) are codified in sequence. TEs are highly abundant, mobile elements of genomic sequence that represent specific evolutionary trajectories within organisms (Hayward & Gilbert, 2022; Wells & Feschotte, 2020). Due to their ability to move within genomes, TEs drive genomic plasticity and have been identified as key players in the evolution of genomic complexity (Schrader & Schmitz, 2019; Bowen & Jordan, 2002). TEs can influence gene expression and regulation by acting as alternative promoters (Faulkner et al., 2009), providing transcription factor binding sites (Sundaram et al., 2014), introducing alternative splicing (Shen et al., 2011), and mediating epigenetic modifications (Drongitis et al., 2019). As such, TEs have also been implicated in disease pathogenesis (Jönsson et al., 2020; Hancks & Kazazian, 2016). Overall, TEs represent a powerful force in evolutionary biology, continually shaping the genetic landscape.

A variety of TEs exist across genomes and can be categorized into several subclasses. The genetic structure of TE types follows regular patterns of structural features and motifs, and thus represents an interesting learning opportunity for sequence models. The Transposable Elements Benchmark (TEB) presents a novel resource for investigating TEs, which represent an area of genome organization that remains underexplored in the genomics deep learning literature. TEB surveys several different TE classes across plant and human genomes. Specifically, TEB offers binary classification datasets for identifying seven specific elements across three different TE classes: retrotransposons, DNA transposons, and pseudogenes. Detailed data preprocessing and dataset statistics are further presented in A.3.

**Genome Understanding Evaluation.** The Genome Understanding Evaluation (GUE) benchmark is a recently published tool that contains seven biologically significant genome analysis tasks that span 28 datasets. Designed to scrutinize the capabilities of genome foundation models, GUE prioritizes genomic datasets that are challenging enough to discern differences between models. The datasets contain sequences ranging from 70–1000 base pairs in length and originating from yeast, mouse, human, and virus genomes. Further details can be found in Zhou et al. (2024).

**Genomic Benchmarks.** We utilize the Genomic Benchmarks (GB) resource, which consists of eight separate classification datasets that spotlight regulatory elements across three different model organisms: human, mouse, and roundworm. Datasets were carefully constructed from published data repositories and consist of input sequences of length 200–500, with the exception of the drosophila enhancers stark dataset, in which sequences have a median length of 2,142. Full details on data preprocessing and dataset summary statistics can be found in Grešová et al. (2023). As the human non-tata promoters dataset in GB was compiled using data that was also used to create the promoter detection datasets in GUE (Dreos et al., 2013), we handle this redundancy by only counting non-overlapping datasets when discussing model performance.

## 5 EXPERIMENTS

### 5.1 GENOMIC CLASSIFICATION

**Data-Generating Scenarios.** The synthetic dataset experiments offer deeper insight into how the hyperbolic inductive bias operates in a genomic learning context. Table 1 suggests that this bias is particularly beneficial in Scenario C, where it aids in uncovering the underlying phylogenetic tree structure in the presence of noise. While this learning mechanism may also help disentangle distinct evolutionary patterns (Scenario B), the results further indicate that discernment in Scenario C may be unrelated to discernment in Scenario A. This distinction likely arises because sequence differentiation occurs at the tree level rather than the clade level.

In evaluating predictive models for biological sequence data, homology splitting is commonly used to assess a model's ability to generalize by excluding homologous sequences. We investigate how this partitioning impacts HCNNs by measuring their zero-shot capability in distinguishing sequences from an unseen phylogenetic tree against randomly sampled background sequences. This experiment, detailed in A.7, demonstrates that hyperbolic models outperform Euclidean models in generalizing to unseen homology branches. These findings suggest that the inductive biases of hyperbolic models offer an even greater advantage than previously estimated, as most genomic datasets overlook this effect and may thus overestimate the performance of predictive methods (Teufel et al., 2023).

**Classification Tasks.** The results from the three classification benchmarks are summarized in Table 2. Across the 42 distinct datasets, the hyperbolic models outperform the equivalent Euclidean model on 37 tasks, as measured by the Matthews correlation coefficient (MCC). In 29 of these datasets, the improvement in score by a hyperbolic model is statistically significant when accounting for variance across different model initializations, whereas the Euclidean CNN statistically outperforms HCNNs in only two datasets.

Further examination of the results suggests that HCNNs confer a particularly strong advantage in distinguishing transcription factor binding sites, epigenetic marks, and TEs in sequence. Across promoter detec-

Table 1: Model performance (MCC) under different synthetic data-generating scenarios, averaged over five random seeds (mean ± standard deviation). The highest-scoring model is in bold, while † denotes a statistically significant improvement over the opposite geometry model(s) with $p < 0.05$, Wilcoxon rank-sum test.

| Scenario | Sequence | Euclidean CNN | Hyperbolic HCNN-S | Hyperbolic HCNN-M |
|---|---|---|---|---|
| **A** | Artificial | 62.38±2.28 | **65.25**±3.27 | 59.25±2.60 |
| | Real | 61.72±3.08 | **66.44**±3.14 | 61.26±2.99 |
| **B** | Artificial | 58.50±0.82 | **60.53**±0.80 | 59.75±0.54 |
| | Real | 57.50±0.88 | **62.53**±6.94 | 59.12±0.54 |
| **C** | Artificial | 62.05±1.62 | **67.65**±1.09 † | 67.43±1.57 † |
| | Real | 66.22±0.44 | **73.62**±0.62 † | 69.30±2.34 † |

tion tasks, hyperbolic embeddings provide no apparent benefit. Since promoters likely function through more complex combinatorial interactions, these dynamics may be more challenging for HCNNs to effectively represent. HCNNs also seem to be significantly disadvantaged in the Covid variant prediction task, which requires distinguishing nine different COVID variants based on their sequences. These findings appear consistent with the synthetic dataset results: the most significant performance gains are observed in scenarios where an evolutionary signal (e.g., transcription factor binding sites, epigenetic marks, TEs) is distinguished from background noise (e.g., non-functional or background sequences). In contrast, the Covid task closely resembles Scenario A, in which a single ancestral sequence evolves along multiple paths.

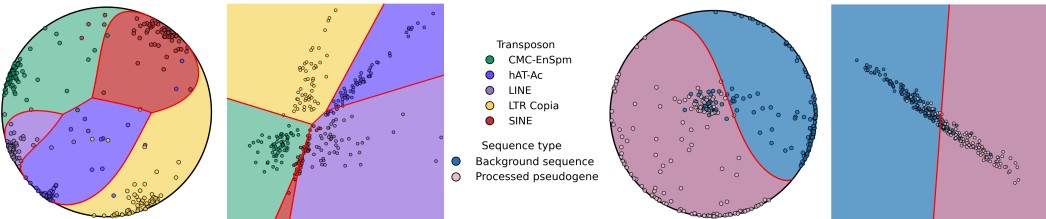

Figure 3: Decision boundaries learned by 2-dimensional HCNNs (circles) and CNNs (squares) for differentiating genomic sequence classes. Boundaries for transposon sequences and processed pseudogenes are visualized on the Poincaré disk and Euclidean plane. Regions are colored by predicted class labels, while points are colored based on their true class labels.

Notably, when comparing the best scoring model across runs, HCNNs outperform DNA language models (LMs) in seven of the 28 GUE datasets (A.4 and Appendix Table 5). Across the majority of tasks, HCNNs outpace HyenaDNA (Nguyen et al., 2024), Caduceus-Ph (Schiff et al., 2024), DNABERT (5-mer), DNABERT (6-mer) (Ji et al., 2021), NT-500M human, NT-500M-1000g, and NT-25000M-1000g (Dalla-Torre et al., 2024). The performance gap between HCNNs and the aforementioned LMs is especially striking given the immense scale of these LMs, which contain $1.7\times$ to $543\times$ more trainable parameters than HCNNs and have undergone pretraining on the entire human genome and 1000 Genomes Project sequences (Byrska-Bishop et al., 2022). HCNNs appear to have a consistent advantage over Euclidean models across many of the core deep learning genomics tasks.

**Expressive Power.** By directly comparing the embeddings and decision boundaries learned by each class of model, we can begin to infer differences in their expressiveness. Figure 3 visualizes the distinctive class boundaries and sequence relationships learned by HCNNs and CNNs, following the setup in (Chlenski et al., 2024). We observe far better separation of classes in the hyperbolic embeddings than in the Euclidean case, lending further credence to the appropriateness of hyperbolic embeddings in a genomic setting.

Additional experiments, detailed in A.10, develop intuition for factors informing the positioning of genomic sequence embeddings in the latent space.

**Embedding Dimensionality.** Prior work on hyperbolic neural networks has demonstrated that the effectiveness of hyperbolic embeddings is especially pronounced in lower dimensions (Chami et al., 2020b; Chamberlain et al., 2017). We probed whether this trend holds under our study conditions by varying the number of channels in the convolutional blocks in both the CNNs and HCNNs. Each distinct model was then trained and evaluated on TEB. The results in Appendix Figure 5 show that HCNN-S exhibits a marginal increase in improvement over CNNs at lower channel dimensions and HCNN-M shows no gains.

Next, we evaluated the potential of the HCNN model class as a foundational framework for DNA LMs. To align more closely with the parameter scales of DNA LMs, we expanded HCNN-S and HCNN-M. Benchmarking these larger models against the two leading model classes that achieve state-of-the-art (SOTA) performance on GUE, DNABERT-2 (Zhou et al., 2024) and NT-2500-multi, suggests that the hyperbolic framework holds promise for DNA LM adoption. As shown in Appendix Table 6, despite having fewer parameters than their competitors, the larger HCNNs achieve SOTA performance on 12 GUE datasets—outperforming DNABERT-2 (11 datasets) and NT-2500-multi (five datasets).

**Learned Curvature.** The curvature of the hyperbolic manifold is a learnable parameter. Exploration of this parameter in TEB (detailed in A.5) illustrates that the value of $K$ does not deviate significantly from its default initialization value of $-1$. However, the HCNN-S and HCNN-M models gravitate towards different curvature values ($K > -1$ and $K < -1$, respectively), with small adjustments in the curvature of the embedding spaces for each block of the model.

**Hybrid Models.** We construct hybrid models that combine Lorentzian and Euclidean components (see A.8 for details). Our results indicate that Euclidean embeddings may still benefit from hyperbolic decision boundaries.

## 5.2 $\delta$-HYPERBOLICITY ESTIMATION

As presented in Appendix Figure 10, our investigation reveals several notable characteristics of $\delta$-hyperbolicity values in finite datasets. The $\delta$ (Appendix Figure 10) and $\delta_{\text{worst}}$ (Appendix Table 9) values computed from the final embedding layer are ostensibly hyperbolic; all values are closer to 0 than 1, indicating tree-like tendencies. However, we observe that the increase in values of $\delta_{\text{worst}}$ are only weakly anticorrelated with relative improvements in performance on learning tasks ($r_S = -0.35, r_M = -0.21$, Appendix Figure 11). An outlier to this pattern appears to be the Covid dataset, which has low hyperbolicity and poor performance from HCNNs.

Table 2: Model performance (MCC) on all real-world genomics datasets, averaged over five random seeds (mean $\pm$ standard deviation). The highest-scoring model is in bold, while $\dagger$ denotes a statistically significant improvement over the opposite geometry model(s) with $p < 0.05$, Wilcoxon rank-sum test. *Note*: the GB human non-tata promoters dataset and GUE promoter detection datasets overlap.

| Benchmark | Task | Dataset | Euclidean CNN | Hyperbolic HCNN-S | Hyperbolic HCNN-M |
|---|---|---|---|---|---|
| TEB | Retrotransposons | LTR Copia | $54.73_{\pm1.45}$ | $64.58_{\pm3.07}$ $\dagger$ | $\mathbf{68.05}_{\pm2.80}$ $\dagger$ |
| | | LINEs | $70.63_{\pm1.24}$ | $76.12_{\pm2.16}$ $\dagger$ | $\mathbf{77.10}_{\pm2.92}$ $\dagger$ |
| | | SINEs | $85.15_{\pm1.64}$ | $\mathbf{85.45}_{\pm1.16}$ | $81.85_{\pm2.95}$ |
| | DNA transposons | CMC-EnSpm | $72.18_{\pm0.32}$ | $\mathbf{80.98}_{\pm1.48}$ $\dagger$ | $80.65_{\pm1.30}$ $\dagger$ |
| | | hAT-Ac | $87.45_{\pm0.90}$ | $89.61_{\pm1.34}$ | $\mathbf{91.04}_{\pm1.58}$ $\dagger$ |
| | Pseudogenes | processed | $60.66_{\pm0.82}$ | $\mathbf{68.30}_{\pm0.93}$ $\dagger$ | $65.41_{\pm5.54}$ |
| | | unprocessed | $51.94_{\pm2.69}$ | $56.13_{\pm0.56}$ $\dagger$ | $\mathbf{58.36}_{\pm1.80}$ $\dagger$ |
| GUE | Epigenetic Marks Prediction | H3 | $64.83_{\pm2.17}$ | $68.14_{\pm1.44}$ | $\mathbf{68.32}_{\pm2.12}$ $\dagger$ |
| | | H3K14ac | $34.27_{\pm6.14}$ | $\mathbf{50.37}_{\pm8.14}$ $\dagger$ | $45.69_{\pm1.95}$ $\dagger$ |
| | | H3K36me3 | $43.74_{\pm2.32}$ | $\mathbf{53.28}_{\pm1.94}$ $\dagger$ | $43.41_{\pm2.00}$ |
| | | H3K4me1 | $28.76_{\pm3.00}$ | $\mathbf{40.84}_{\pm1.18}$ $\dagger$ | $34.71_{\pm3.70}$ |
| | | H3K4me2 | $25.38_{\pm5.40}$ | $\mathbf{39.74}_{\pm4.61}$ $\dagger$ | $29.53_{\pm1.97}$ |
| | | H3K4me3 | $21.77_{\pm5.58}$ | $\mathbf{49.51}_{\pm0.96}$ $\dagger$ | $30.39_{\pm3.32}$ $\dagger$ |
| | | H3K79me3 | $54.88_{\pm2.09}$ | $\mathbf{62.39}_{\pm2.14}$ $\dagger$ | $58.48_{\pm1.88}$ |
| | | H3K9ac | $40.37_{\pm3.89}$ | $\mathbf{52.90}_{\pm1.12}$ $\dagger$ | $50.21_{\pm1.52}$ $\dagger$ |
| | | H4ac | $31.59_{\pm8.45}$ | $\mathbf{52.29}_{\pm0.93}$ $\dagger$ | $44.88_{\pm4.70}$ |
| | | H4 | $74.81_{\pm0.92}$ | $75.43_{\pm1.49}$ | $\mathbf{76.20}_{\pm0.61}$ |
| | Human Transcription Factor Prediction | 0 | $58.65_{\pm3.40}$ | $\mathbf{62.84}_{\pm0.64}$ | $60.92_{\pm1.72}$ |
| | | 1 | $61.41_{\pm1.60}$ | $\mathbf{67.13}_{\pm2.59}$ $\dagger$ | $66.76_{\pm1.25}$ $\dagger$ |
| | | 2 | $49.79_{\pm0.51}$ | $67.17_{\pm5.26}$ $\dagger$ | $\mathbf{68.36}_{\pm2.70}$ $\dagger$ |
| | | 3 | $35.67_{\pm0.30}$ | $41.96_{\pm2.95}$ | $\mathbf{42.93}_{\pm2.30}$ $\dagger$ |
| | | 4 | $57.68_{\pm0.26}$ | $66.01_{\pm1.88}$ $\dagger$ | $\mathbf{67.99}_{\pm2.30}$ $\dagger$ |
| | Splice Site Prediction | reconstructed | $78.64_{\pm0.43}$ | $80.32_{\pm1.24}$ $\dagger$ | $\mathbf{80.76}_{\pm1.06}$ $\dagger$ |
| | Mouse Transcription Factor Prediction | 0 | $22.51_{\pm2.78}$ | $46.09_{\pm2.17}$ $\dagger$ | $\mathbf{47.96}_{\pm5.01}$ $\dagger$ |
| | | 1 | $76.56_{\pm0.51}$ | $\mathbf{78.93}_{\pm0.31}$ $\dagger$ | $76.68_{\pm0.81}$ |
| | | 2 | $62.69_{\pm1.52}$ | $74.76_{\pm3.07}$ $\dagger$ | $\mathbf{74.78}_{\pm2.98}$ $\dagger$ |
| | | 3 | $36.93_{\pm8.35}$ | $\mathbf{68.61}_{\pm4.24}$ $\dagger$ | $66.58_{\pm3.24}$ $\dagger$ |
| | | 4 | $30.23_{\pm3.13}$ | $40.07_{\pm0.83}$ $\dagger$ | $\mathbf{40.57}_{\pm2.09}$ $\dagger$ |
| | Covid Variant Classification | Covid | $\mathbf{66.43}_{\pm0.48}$ $\dagger$ | $36.71_{\pm9.69}$ | $14.81_{\pm0.46}$ |
| | Core Promoter Detection | tata | $78.26_{\pm2.85}$ | $79.54_{\pm1.61}$ | $\mathbf{79.87}_{\pm2.50}$ |
| | | notata | $\mathbf{66.60}_{\pm1.07}$ | $66.52_{\pm0.28}$ | $65.95_{\pm0.51}$ |
| | | all | $66.47_{\pm0.74}$ | $65.26_{\pm1.11}$ | $\mathbf{67.16}_{\pm0.55}$ |
| | Promoter Detection | tata | $78.58_{\pm3.39}$ | $\mathbf{79.74}_{\pm2.66}$ | $78.77_{\pm0.78}$ |
| | | notata | $\mathbf{90.81}_{\pm0.51}$ | $89.86_{\pm0.76}$ | $90.28_{\pm0.37}$ |
| | | all | $\mathbf{88.00}_{\pm0.39}$ | $87.60_{\pm0.51}$ | $87.93_{\pm0.76}$ |
| GB | Demo | coding vs intergenomic seqs | $75.14_{\pm0.35}$ | $80.04_{\pm0.28}$ $\dagger$ | $\mathbf{80.25}_{\pm0.24}$ $\dagger$ |
| | | human or worm | $89.89_{\pm0.15}$ | $92.65_{\pm0.11}$ $\dagger$ | $\mathbf{92.71}_{\pm0.27}$ $\dagger$ |
| | Enhancers | drosophila enhancers stark | $7.99_{\pm3.01}$ | $10.77_{\pm2.34}$ | $\mathbf{10.87}_{\pm3.32}$ |
| | | human enhancers cohn | $30.76_{\pm2.05}$ | $46.63_{\pm0.88}$ $\dagger$ | $\mathbf{46.68}_{\pm1.11}$ $\dagger$ |
| | | human enhancers ensembl | $\mathbf{79.48}_{\pm0.10}$ $\dagger$ | $44.48_{\pm2.94}$ | $72.99_{\pm0.36}$ |
| | Regulatory | human ensembl regulatory | $89.73_{\pm0.21}$ | $89.91_{\pm0.72}$ | $\mathbf{90.21}_{\pm1.37}$ |
| | | human non-tata promoters* | $64.98_{\pm0.21}$ | $\mathbf{83.57}_{\pm0.73}$ $\dagger$ | $79.90_{\pm1.48}$ $\dagger$ |
| | Open Chromatin Regions | human ocr ensembl | $39.92_{\pm0.85}$ | $\mathbf{56.22}_{\pm0.28}$ $\dagger$ | $55.36_{\pm2.52}$ $\dagger$ |

Previous studies have attempted to calibrate their reported $\delta_{\text{worst}}$ values by comparing them to empirical estimates of $\delta_{\text{worst}}$ for the Poincaré disk $\mathbb{D}^2$, and the 2-sphere $S^2$ (Khrulkov et al., 2020; Yang et al., 2024), however we note that these empirical estimates are for metric spaces that are categorically much lower in dimensionality than the feature spaces used for the dataset embeddings, leading to potentially incongruous comparisons. Indeed, we find that high-dimensional data produces "emergent hyperbolicity", with points at higher dimensions producing smaller $\delta_{\text{worst}}$ and $\delta_{\text{avg}}$ values (detailed in A.9.2). Our results highlight a pronounced disparity: the difference in empirical $\delta$ values between embeddings sampled on $\mathbb{H}^2$ and those sampled on higher-dimensional hyperbolic spaces ($\mathbb{H}^d$, where $d \in [200, 1000]$) – with comparable magnitudes to the sequence embeddings – can be as large as 0.2 (Appendix Figure 12). This disparity becomes even more pronounced on Euclidean ($\mathbb{R}^d$) and hyperspherical ($\mathbb{S}^d$) manifolds. Such significant differences in $\delta$ values may largely determine whether the estimated $\delta$ indicates a more hyperbolic nature of the underlying space or otherwise.

To provide a more equitable calibration of hyperbolicity, we compare the $\delta$ distributions from our genomic datasets to those from simulated datasets of matching dimensionality. We generate these simulated datasets on both Euclidean and hyperbolic ($K = -1$) manifolds. Appendix Figure 10 illustrates the $\delta$ distributions for each set of dataset embeddings, where each embedding $G \in \mathbb{R}^{d_F}$, with $d_F$ as the final embedding layer size. Our results reveal that the majority of the genomic dataset embeddings exhibit greater hyperbolicity (lower $\delta$ values) compared to embeddings simulated from a baseline Gaussian distribution on a Euclidean manifold of the same dimensionality. To quantify this difference, we employ the Wilcoxon rank-sum test between the baseline and the genome dataset distributions. This analysis shows that 25 out of 43 sequence datasets have significantly lower $\delta$ values than the baseline ($p < 0.05$). These findings support the hypothesis that genomic sequence data may possess an innate hyperbolicity, making them better suited to hyperbolic representations.

Our approach of examining the entire distribution of $\delta$ values, rather than relying on a single scalar measure, reveals nuanced insights into the hyperbolic tendencies of different datasets. This comprehensive view allows us to capture subtleties that might otherwise be overlooked. For instance, the H3K36me3 dataset exhibits a $\delta$ distribution that is significantly lower in hyperbolicity compared to the baseline. However, its high $\delta_{\text{worst}}$ estimate suggests that it may be less hyperbolic than the baseline when considering only this single metric. Similarly, while the TEB datasets show relatively large $\delta_{\text{worst}}$ estimates, their $\delta$ distributions are notably right-skewed. These characteristics appear more consistent with the superior performance of HCNN models on these datasets.

The discrepancies between single-point estimates ($\delta_{\text{worst}}$, $\delta_{\text{avg}}$) and full distributions underscore the importance of a more holistic approach. By considering the entire spectrum of $\delta$ values across the feature space, we gain a more accurate characterization of the data's tree-like properties. This comprehensive perspective not only provides a richer understanding of the dataset's geometric structure but also offers better insights into why models like HCNNs perform well. Expanding this analysis to DNA LMs (Section A.9.3) reveals that these characteristics generalize across a broader range of models. Still, while moving beyond scalar metrics and calibrating against dimensionality-matched geometries has uncovered hyperbolic tendencies in genomic data that point estimates miss, critical challenges persist: formalizing the behavior of $\delta$-distributions statistically, particularly in the face of emergent hyperbolicity, and exploring their robustness to the choice of metric would significantly clarify in which situations hyperbolic representation learning is applicable.

## 6 CONCLUSION

We introduce a novel application of HCNNs for genomic sequence modeling, critically evaluating their strengths and limitations. Our findings show that hyperbolic embeddings offer a distinct performance advantage in key genomics tasks, particularly under resource constraints. Additionally, our analysis of dataset embeddings uncovers significant correlations between dimensionality and $\delta$-hyperbolicity, reinforcing the value of hyperbolic space for genome representation.

HCNNs are lightweight, modular models with the scalability to produce competitive DNA LMs, offering additional performance gains through pretraining and complementary techniques. Moreover, this work drives future research toward developing robust metrics for evaluating dataset hyperbolicity and formalizing its relationship with curvature and dimensionality. By advancing the understanding and optimization of hyperbolic models in genomics, our study encourages deeper exploration of this promising paradigm.

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

# A    APPENDIX

## A.1    LORENTZ CONVOLUTIONAL LAYER

### A.1.1    LAYER COMPONENTS

We further break down the Lorentz convolutional layer by defining each separate transformation. First, given hyperbolic points $\{\boldsymbol{x}_i\}_{i=1}^N$, the Lorentz direct concatenation (HCat) (Qu & Zou, 2022) is defined as:

$$\boldsymbol{y} = \mathrm{HCat}(\{\boldsymbol{x}_i\}_{i=1}^N) = \left[ \sqrt{\sum_{i=1}^N x_{i_t}^2 + \frac{N-1}{K}}, \boldsymbol{x}_{1_s}^T, \ldots, \boldsymbol{x}_{N_s}^T \right]^T \tag{12}$$

with $\boldsymbol{y} \in \mathbb{L}_K^{nN} \subset \mathbb{R}^{nN+1}$. This manipulation provides a numerically stable way to concatenate hyperbolic representations. Next, Chen et al. (2022b) introduced a Lorentz fully-connected layer. Given the input vector $\boldsymbol{x}$ and the weight parameters $\mathbf{W} \in \mathbb{R}^{m \times n+1}, \boldsymbol{v} \in \mathbb{R}^{n+1}$ for the fully connected layer, the transformation matrix is defined as:

$$f_{\boldsymbol{x}}\left( \begin{bmatrix} \boldsymbol{v}^T \\ \mathbf{W} \end{bmatrix} \right) = \begin{bmatrix} \frac{\sqrt{\|\mathbf{W}\boldsymbol{x}\|^2 - 1/K}}{\boldsymbol{v}^T \boldsymbol{x}} \boldsymbol{v}^T \\ \mathbf{W} \end{bmatrix}. \tag{13}$$

Then, adding in other layer components (except for internal layer normalization) results in the following formula:

$$\boldsymbol{y} = \mathrm{LFC}(\boldsymbol{x}) = \left[ \sqrt{\|\psi(\mathbf{W}\boldsymbol{x} + \mathbf{b})\|^2 - 1/K} \\ \psi(\mathbf{W}\boldsymbol{x} + \mathbf{b}) \right] \tag{14}$$

where $\mathbf{b} \in \mathbb{R}^n$ and $\psi$ denote the bias and activation, respectively.

### A.1.2    LAYER MAPPING

HCNN-M models leverage multiple manifolds with corresponding curvatures $[K_1, ..., K_u]$ for each of $u$ designated blocks. Therefore, we define the mapping between manifolds as follows, using the definitions of exponential and logarithmic maps defined in equations 3 and 4, respectively. For a mapping of point $\mathbf{x} \in \mathcal{M}_1$ (where $\mathcal{M}_1$ has corresponding curvature $K_1$) to the manifold $\mathcal{M}_2$ (with curvature $K_2$), we must first apply a logarithmic map at the origin to bring $\mathbf{x}$ to the tangent space $T_{\mathbf{0}}\mathcal{M}_1$. Then, we apply an exponential map at the origin of the resulting point to the new manifold $\mathcal{M}_2$. The layer map operation $\mathbf{LM}_{\mathcal{M}_1 \rightarrow \mathcal{M}_2}(\mathbf{x})$ can therefore be defined as follows:

$$\mathbf{LM}_{\mathcal{M}_1 \rightarrow \mathcal{M}_2}(\mathbf{x}) = \exp_{\mathbf{0}}^{K_2}(\log_{\mathbf{0}}^{K_1}(\mathbf{x})). \tag{15}$$

## A.2    MODELING DETAILS

### A.2.1    MODEL

A detailed breakdown of the CNN/HCNN model architecture is visualized in Figure 4. The HCNNs use the Lorentz formulation of each model component. For HCNN-M, we show the partition of each manifold across each segment of the architecture. We use cross-entropy loss as our objective and train each model end-to-end on each dataset.

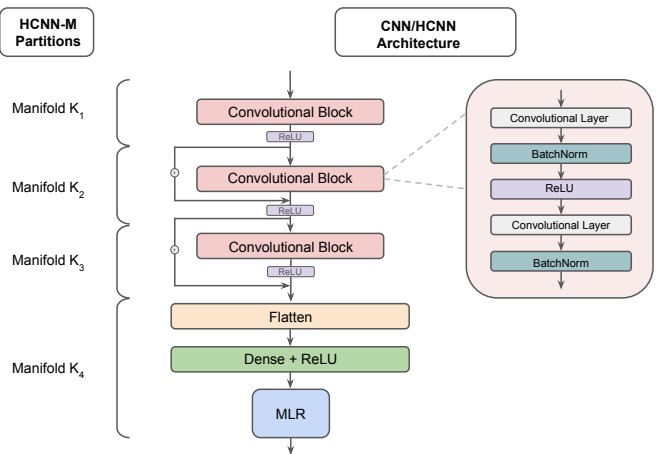

Figure 4: The generalized block architecture for the CNNs/HCNNs. On the left, we delineate the manifold partitions used in our HCNN-M models.

### A.2.2 HYPERPARAMETERS

When possible, we keep the hyperparameters constant across the different model types (Table 3). However, we train the Euclidean CNN using the AdamW optimizer (Loshchilov & Hutter, 2019) and the HCNNs using RiemannianAdam (Bécigneul & Ganea, 2019).

Table 3: Hyperparameter settings for CNN/HCNN training.

|  | Euclidean CNN | HCNN-S | HCNN-M |
|---|---|---|---|
| Optimizer | AdamW | RiemannianAdam | RiemannianAdam |
| Learning Rate (TEB/GUE/GB) | 1e-4, 1e-4, 1e-5 | 1e-4, 1e-4, 1e-5 | 1e-4, 1e-4, 1e-5 |
| Manifold Learning Rate | N/A | 2e-2 | 2e-2 |
| Batch size | 100 | 100 | 100 |
| Weight decay | 0.1 | 0.1 | 0.1 |
| Epochs | 100 | 100 | 100 |
| $\beta_1, \beta_2$ | 0.9, 0.999 | 0.9, 0.999 | 0.9, 0.999 |

### A.3 TRANSPOSABLE ELEMENTS BENCHMARK

TEB presents seven distinct sequence classification datasets categorized within three prediction tasks. An overview of the datasets is presented in Table 4. Sequence and annotation data were integrated from both human and plant genome datasets.

For the retrotransposon and DNA transposon tasks, we crafted a dataset by employing annotations from PlantRep (Luo et al., 2022), a database that provides comprehensive annotations of plant repetitive elements across 459 plant genomes. We narrowed the number of candidate species to those that had an appropriate number of TEs of interest to power deep learning tasks, as well as an average TE sequence length of similar magnitude to the other benchmark datasets (200-1000 bp). Then, we randomly selected *Oryza glumipatula* from the set of candidate species to use as the plant species for our benchmark. Annotations were downloaded from PlantRep, while the *Oryza glumipatula* genome (v1.5) was downloaded from the NCBI genome browser (https://ftp.ncbi.nlm.nih.gov). Within the retrotransposon group, there are three datasets: LTR Copia, LINEs, and SINEs. LTR Copia are a type of retrotransposon characterized by a pair of identical flanking repetitive regions called long terminal repeats (LTRs). Conversely, long interspersed nuclear elements (LINEs), and short interspersed nuclear elements (SINEs) are retrotransposons that do not contain LTRs, and generally contain a promoter while varying by length. Next, within the DNA transposon group, we target two

of the most ubiquitous subfamilies: CMC-EnSpm and hAT-Ac, each of which is distinguished by specific short terminal inverted repeats.

While pseudogenes themselves are not a type of TE, they are often the result of TE activity. Therefore, we examine the presence of pseudogenes in the human reference genome (GRCh38.p12), using gene/transcript biotype annotations from GENCODE and Ensembl (Frankish et al., 2019). Pseudogenes are classified as processed and unprocessed, each of which results from a different mechanism of action. A processed pseudogene lacks introns and arises from reverse transcription of mRNA followed by reinsertion of DNA into the genome, while an unprocessed pseudogene may contain introns and is the product of a gene duplication event.

For dataset construction, we created a positive set of sequences spanning each TE of interest. We then generated a negative set by randomly sampling non-overlapping, remaining portions of the genome (without replacement) until we had a matching number of negative sequences. We used a chromosome level train/validation/test split for our sequences, separating out chromosomes 8/9 and 20-22/17-19 for validation/test sets in *Oryza glumipatula* and human, respectively, while the remaining chromosomes were used for the training sets.

Table 4: Summary statistics for TEB, including the specific type of TE and the number of training, validation, and test samples in each dataset.

| Prediction Task | Species | Max Length | Datasets | Train / Dev / Test |
|---|---|---|---|---|
| **Retrotransposons** | Plant | 500 1000 500 | LTR Copia LINEs SINEs | 7666 / 682 / 568 22502 / 2030 / 1782 21152 / 1836 / 1784 |
| **DNA Transposons** | Plant | 200 1000 | CMC-EnSpm hAT-Ac | 19912 / 1872 / 1808 17322 / 1822 / 1428 |
| **Pseudogenes** | Human | 1000 1000 | processed unprocessed | 17956 / 1046 / 1740 12938 / 766 / 884 |

## A.4 DNA LANGUAGE MODELS

We compare the best classification performance of our HCNN models to that of several DNA LMs. Table 5 documents the performance of ten large DNA LMs on the GUE datasets, along with the number of trainable parameters present in each model. We benchmarked HyenaDNA and Caduceus-Ph, while for other models, we used the benchmarking results reported in (Zhou et al., 2024). For the HCNN-S and HCNN-M models, we report the average number of model parameters used across all GUE datasets. Below, we provide a short description of each model class:

- *HyenaDNA*: A long-context DNA LM that uses the Hyena operator as a basic building block (Poli et al., 2023), which is a subquadratic alternative to attention. HyenaDNA utilizes extended convolutions and data-controlled gating mechanisms to identify long-range genomic effects (Nguyen et al., 2024).

- *Caduceus-Ph*: A bidirectional DNA LM for long-range sequence modeling that builds on the Mamba module (Gu & Dao, 2024; Schiff et al., 2024).

- *DNABERT (5-mer, 6-mer)*: An early iteration of a pretrained transformer model for the genome, DNABERT (Ji et al., 2021) uses the BERT architecture and is trained on human DNA sequences. There are four variants of the model, and here we list the results for the 5-mer and 6-mer versions, which use overlapping 5-mer and 6-mer tokenization of sequences.

- *Nucleotide Transformer (500M human, 500M 1000g, 2500M 1000g, 2500M multi)*: Nucleotide Transformer (NT) represents the largest class of models in terms of parameters and training data. There are four variants of NT. The labels "500M" and "2500M" correspond to the number of trainable parameters in the model (Dalla-Torre et al., 2024). For the training data, the categories "human", "1000g", and "multi" refer to the human reference genome,

the 3203 human genomes from the 1000 Genome project, and genomes from 850 different species, respectively.

- *DNABERT-2, DNABERT-2-PT*: A refinement of DNABERT, DNABERT-2 incorporates Byte-Pair Encoding and several architectural upgrades for improved learning capabilities. DNABERT-2 is pretrained on the human reference genome, whereas DNABERT-2-PT is further pretrained on the training sets of the 28 GUE datasets (Zhou et al., 2024).

Table 5: The performance (F1-score for Covid, MCC for all other datasets) of several prominent DNA LMs in comparison to the HCNNs on GUE. The best-performing score for each GUE dataset is bolded.

| | Caduceus -Ph | Hyena DNA | DNA BERT (5-mer) | DNA BERT (6-mer) | NT -500M human | NT -500M 1000g | NT -2500M 1000g | NT -2500M multi | DNA BERT-2 | DNA BERT-2 -PT | HCNN -S | HCNN -M |
|---|---|---|---|---|---|---|---|---|---|---|---|---|
| Parameters | 7.7M | 28.2M | 87M | 89M | 500M | 500M | 2.5B | 2.5B | 117M | 117M | 4.6M | 4.6M |
| H3 | 77.09 | 67.17 | 73.40 | 73.10 | 69.67 | 72.52 | 74.61 | 78.77 | 78.27 | **80.17** | 69.42 | 69.95 |
| H3K14ac | 41.44 | 31.98 | 40.68 | 40.06 | 33.55 | 39.37 | 44.08 | **56.20** | 52.57 | 57.42 | 56.03 | 48.25 |
| H3K36me3 | 46.49 | 48.27 | 48.29 | 47.25 | 44.14 | 45.58 | 50.86 | **61.99** | 56.88 | 61.90 | 55.27 | 45.76 |
| H3K4me1 | 37.76 | 35.83 | 40.65 | 41.44 | 37.15 | 40.45 | 43.10 | **55.30** | 50.52 | 53.00 | 41.86 | 39.78 |
| H3K4me2 | 28.16 | 25.81 | 30.67 | 32.27 | 30.87 | 31.05 | 30.28 | 36.49 | 31.13 | 39.89 | **43.88** | 31.27 |
| H3K4me3 | 24.40 | 23.15 | 27.10 | 27.81 | 24.06 | 26.16 | 30.87 | 40.34 | 36.27 | 41.20 | **50.58** | 33.59 |
| H3K79me3 | 60.31 | 54.09 | 59.61 | 61.17 | 58.35 | 59.33 | 61.20 | 64.70 | **67.39** | 65.46 | 64.62 | 63.35 |
| H3K9ac | 52.70 | 50.84 | 51.11 | 51.22 | 45.81 | 49.29 | 52.36 | 56.01 | 55.63 | **57.07** | 54.09 | 52.25 |
| H4 | 79.91 | 73.69 | 77.27 | 79.26 | 76.17 | 76.29 | 79.76 | 81.67 | 80.71 | **81.86** | 77.24 | 76.94 |
| H4ac | 40.90 | 38.44 | 37.48 | 37.43 | 33.74 | 36.79 | 41.46 | 49.13 | 50.43 | 50.35 | **52.94** | 51.86 |
| prom all | 85.87 | 47.38 | 90.16 | 90.48 | 87.71 | 89.76 | 90.95 | **91.01** | 86.77 | 88.31 | 88.23 | 88.83 |
| prom notata | 93.23 | 52.24 | 92.45 | 93.05 | 90.75 | 91.75 | 93.07 | 94.00 | 94.27 | **94.34** | 90.92 | 90.74 |
| prom tata | 66.07 | 5.34 | 69.51 | 61.56 | 78.07 | 78.23 | 75.80 | 79.43 | 71.59 | 68.79 | **82.70** | 79.80 |
| Human TF 0 | 67.32 | 62.30 | 66.97 | 66.84 | 61.59 | 63.64 | 66.31 | 66.64 | **71.99** | 69.12 | 63.56 | 63.35 |
| Human TF 1 | 72.10 | 67.86 | 69.98 | 70.14 | 66.75 | 70.17 | 68.30 | 70.28 | **76.06** | 71.87 | 69.39 | 68.48 |
| Human TF 2 | 58.92 | 46.85 | 59.03 | 61.03 | 53.58 | 52.73 | 58.70 | 58.72 | 66.52 | 62.96 | **73.80** | 71.40 |
| Human TF 3 | 54.85 | 41.78 | 52.95 | 51.89 | 42.95 | 45.24 | 49.08 | 51.65 | **58.54** | 55.35 | 44.08 | 43.66 |
| Human TF 4 | 69.45 | 61.23 | 69.26 | 70.97 | 60.81 | 62.82 | 67.59 | 69.34 | **77.43** | 74.94 | 68.43 | 70.01 |
| c. prom all | 67.28 | 36.95 | 69.48 | 68.90 | 63.45 | 66.70 | 67.39 | **70.33** | 69.37 | 67.50 | 66.33 | 67.84 |
| c. prom notata | 66.07 | 35.38 | 69.81 | 70.47 | 64.82 | 67.17 | 67.46 | **71.58** | 68.04 | 69.53 | 66.78 | 66.48 |
| c. prom tata | 72.94 | 72.87 | 76.79 | 76.06 | 71.34 | 73.52 | 69.66 | 72.97 | 74.17 | 76.18 | 81.34 | **82.07** |
| Mouse TF 0 | 56.18 | 35.62 | 42.45 | 44.42 | 31.04 | 39.26 | 48.31 | 63.31 | 56.76 | **64.23** | 48.41 | 52.31 |
| Mouse TF 1 | 80.31 | 80.50 | 79.32 | 78.94 | 75.04 | 75.49 | 80.02 | 83.76 | 84.77 | **86.28** | 79.26 | 77.41 |
| Mouse TF 2 | 75.89 | 65.34 | 62.22 | 71.44 | 61.67 | 64.70 | 70.14 | 71.52 | 79.32 | **81.28** | 77.86 | 77.51 |
| Mouse TF 3 | 73.47 | 54.20 | 49.92 | 44.89 | 29.17 | 33.07 | 42.25 | 69.44 | 66.47 | 73.49 | **73.51** | 69.73 |
| Mouse TF 4 | 47.98 | 19.17 | 40.34 | 42.48 | 29.27 | 34.01 | 43.40 | 47.07 | **52.66** | 50.80 | 41.27 | 43.62 |
| Covid | 45.19 | 23.27 | 50.46 | 55.50 | 50.82 | 52.06 | 66.73 | **73.04** | 71.02 | 68.49 | 46.43 | 24.74 |
| Splice | 81.59 | 72.67 | 84.02 | 84.07 | 79.71 | 80.97 | 85.78 | **89.35** | 84.99 | 85.93 | 81.96 | 82.23 |

## A.5 MANIFOLD CURVATURE

Figure 6 depicts the learned curvatures for models trained on TEB. In the HCNN-M models, blocks 1-3 represent the hyperbolic convolutional blocks in the model, each associated with a corresponding manifold that has its own curvature. Block 4 represents the portion of the model that involves a flattening step, a dense layer, and MLR, operations that all occur on a single hyperbolic manifold (Figure 4). In the HCNN-S models, the value of $K$ is fixed, as a single manifold is used across the entire model.

## A.6 SYNTHETIC DATASETS

We construct each synthetic dataset by randomly sampling a phylogenetic tree using the Environment for Tree Exploration (ETE) toolkit Huerta-Cepas et al. (2016). To simulate nucleotide sequence evolution along the tree's branches, we use the PYVOLVE package (Spielman & Wilke, 2015), specifically for its implementation of the Generalized Time-Reversible (GTR) model (Tavaré, 1984)

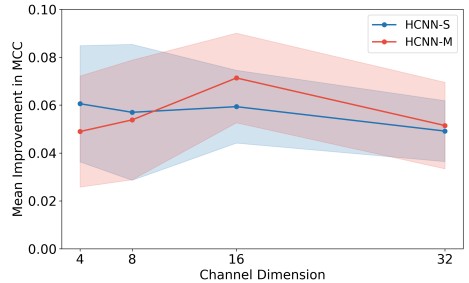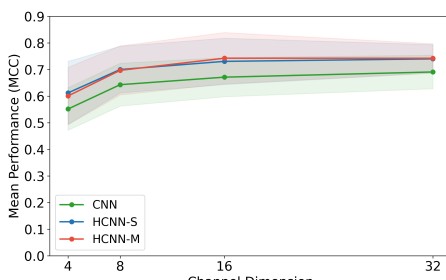

Figure 5: On the left, we show the average improvement in performance (MCC) on TEB datasets for HCNNs compared to CNNs as the channel dimension in the convolutional layers varies. On the right, we present the mean MCC achieved by the models across TEB datasets, for each channel dimension.

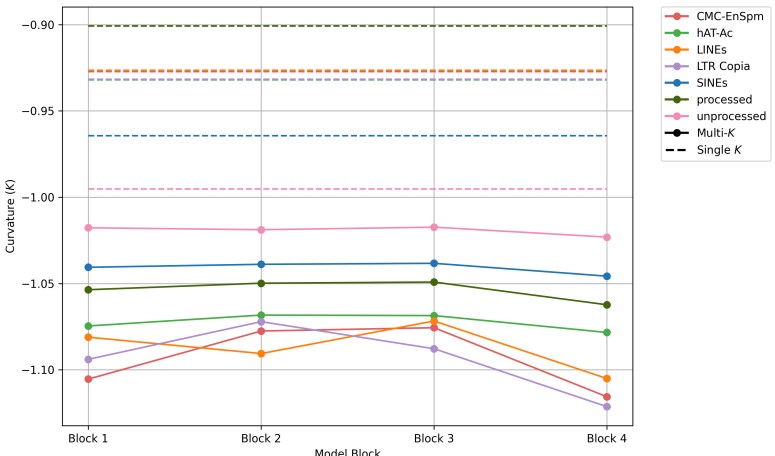

Figure 6: Average values of $K$, the curvature parameter in the HCNNs, as they vary across each block of the model. Values are reported for models trained on each of the seven classification tasks in TEB.

with default parameters. Four types of fixed-length sequences are generated and used across Scenarios A, B, and C:

**Artificial tree**: The starting ancestral (root) sequence is randomly generated.

**Real tree**: The starting ancestral sequence is sampled from the human genome.

**Artificial background sequence**: Sequences are generated randomly and independently by sampling nucleotides.

**Real background sequence**: Sequences are sampled from independent (different chromosome) regions of the human genome relative to the starting ancestral sequence.

We define the task for each scenario as follows:

(A) **Intra-tree differentiation**: A single tree is sampled, with clade membership determining class labels. The model's task is to differentiate clades.

(B) **Inter-tree differentiation**: A different tree (with a different starting ancestral sequence) is sampled for each label. The model's task is to differentiate trees.

(C) **Tree identification**: A single tree is sampled, and all sequences from this tree share the same label. Independently sampled background sequences are assigned a separate label. The model's task is to differentiate the tree from the background sequences.

Table 6: The performance (F1-score for Covid, MCC for all other datasets) of SOTA DNA LMs and scaled HCNNs on GUE benchmark datasets. The best-performing score for each dataset is bolded. For the scaled HCNN-S and HCNN-M models, we report the average number of model parameters used across all GUE datasets.

| | NT -2500M -multi | DNABERT-2 | DNABERT-2 -PT | HCNN-S (Large) | HCNN-M (Large) |
|---|---|---|---|---|---|
| Parameters | 2.5B | 117M | 117M | 43M | 43M |
| H3 | 78.77 | 78.27 | **80.17** | 72.17 | 72.21 |
| H3K14ac | 56.20 | 52.57 | 57.42 | **70.56** | 69.87 |
| H3K36me3 | 61.99 | 56.88 | 61.90 | **68.06** | 67.92 |
| H3K4me1 | 55.30 | 50.52 | 53.00 | 53.33 | **55.45** |
| H3K4me2 | 36.49 | 31.13 | 39.89 | **54.67** | 52.50 |
| H3K4me3 | 40.34 | 36.27 | 41.20 | **67.25** | 64.61 |
| H3K79me3 | 64.70 | 67.39 | 65.46 | 70.49 | **70.65** |
| H3K9ac | 56.01 | 55.63 | 57.07 | **63.36** | 60.66 |
| H4 | 81.67 | 80.71 | **81.86** | 76.54 | 74.78 |
| H4ac | 49.13 | 50.43 | 50.35 | 62.16 | **67.30** |
| promoter all | **91.01** | 86.77 | 88.31 | 83.35 | 83.73 |
| promoter notata | 94.00 | 94.27 | **94.34** | 88.36 | 90.67 |
| promoter tata | 79.43 | 71.59 | 68.79 | **81.86** | 79.19 |
| Human TF 0 | 66.64 | **71.99** | 69.12 | 65.09 | 62.85 |
| Human TF 1 | 70.28 | **76.06** | 71.87 | 67.59 | 69.91 |
| Human TF 2 | 58.72 | 66.52 | 62.96 | **70.73** | 63.79 |
| Human TF 3 | 51.65 | **58.54** | 55.35 | 42.12 | 46.26 |
| Human TF 4 | 69.34 | **77.43** | 74.94 | 71.95 | 70.33 |
| core promoter all | **70.33** | 69.37 | 67.50 | 61.77 | 62.56 |
| core promoter notata | **71.58** | 68.04 | 69.53 | 66.01 | 65.71 |
| core promoter tata | 72.97 | 74.17 | 76.18 | 80.20 | **80.26** |
| Mouse TF 0 | 63.31 | 56.76 | **64.23** | 48.84 | 49.24 |
| Mouse TF 1 | 83.76 | 84.77 | **86.28** | 81.07 | 79.43 |
| Mouse TF 2 | 71.52 | 79.32 | **81.28** | 80.51 | 75.61 |
| Mouse TF 3 | 69.44 | 66.47 | 73.49 | **81.57** | 78.77 |
| Mouse TF 4 | 47.07 | **52.66** | 50.80 | 41.79 | 43.70 |
| Covid | **73.04** | 71.02 | 68.49 | 45.06 | 31.09 |
| Splice | **89.35** | 84.99 | 85.93 | 79.23 | 78.84 |

Representative simulated phylogenetic trees and their corresponding labels are visualized in Figures 7 and 8. We introduce noise into the datasets by randomly swapping $10\%$ of the labels in the training and validation sets.

## A.7 HOMOLOGY SPLITTING

The experimental setup for homology splitting is visualized in Figure 9. For the training and validation data, we generate a synthetic dataset as in Scenario C, where sequences generated from the tree share the same label, and background sequences not originating from the tree share a different label. However, instead of creating a test set from this dataset, we create the test set by generating a completely new phylogenetic tree and sampling sequences from it. The tree-generated sequences in the test dataset thus originate from entirely unseen homology branches.

Results of this experiment are presented in Table 7. Hyperbolic models show significantly improved generalization over the Euclidean model in an evolutionary and phylogenetic context.

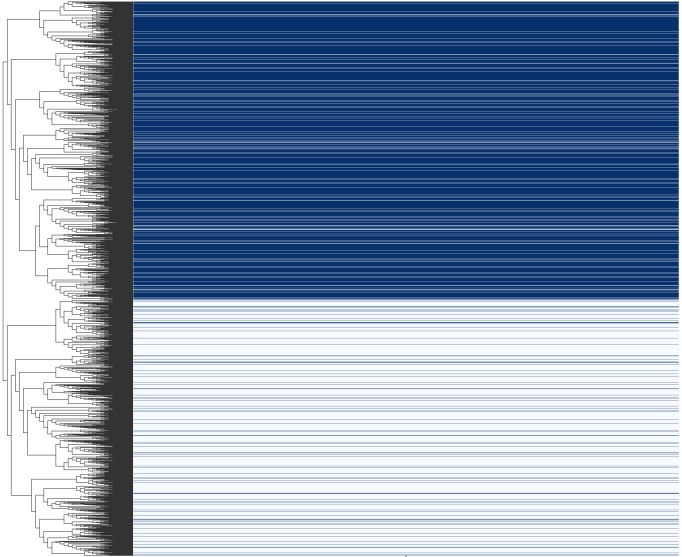

Figure 7: Leaf node sequence classifications (with added noise) in Scenario A for the simulated phylogenetic tree (structure visible on the left).

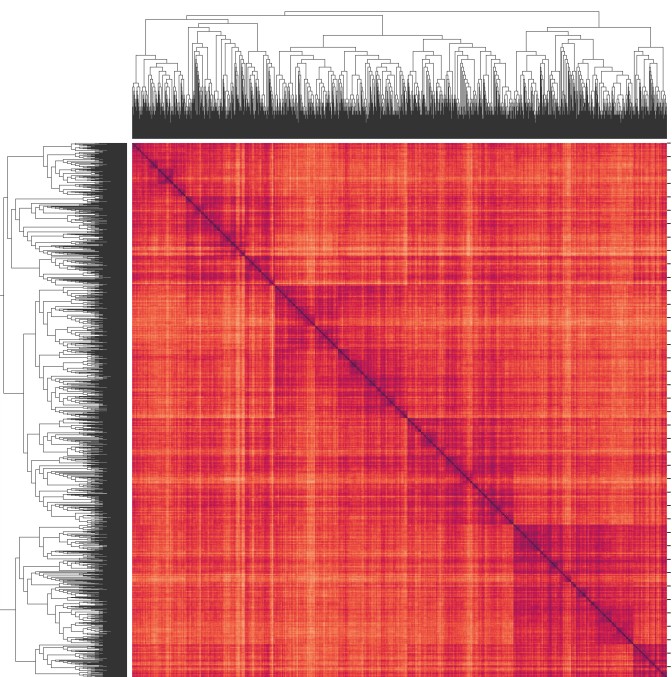

Figure 8: Hamming distance matrix for all leaves in the simulated phylogenetic tree for Scenario A.

Table 7: Model performance (MCC) on the homology splitting experiment, averaged over five random seeds (mean $\pm$ standard deviation). The highest-scoring model is in bold, while $\dagger$ denotes a statistically significant improvement over the opposite geometry model(s) with $p < 0.05$, Wilcoxon rank-sum test.

| CNN | HCNN-S | HCNN-M |
|---|---|---|
| $24.31_{\pm 7.99}$ | $\mathbf{45.73}_{\pm 8.93}{}^{\dagger}$ | $40.87_{\pm 8.93}{}^{\dagger}$ |

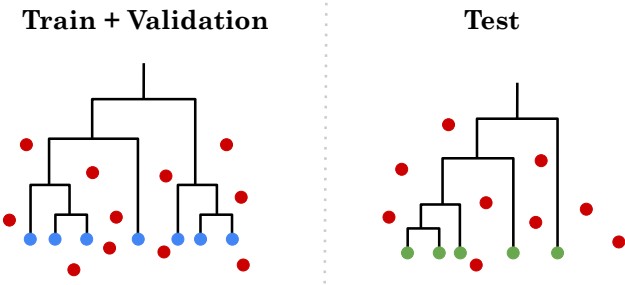

Figure 9: Overview of the homology splitting experiment. A training and validation dataset (left) are generated in the same manner as the synthetic dataset used for Scenario C. For the test dataset (right), a completely new tree and ancestral sequence are used to generate the tree class.

## A.8    HYBRID MODELS

Following Bdeir et al. (2024), we evaluate hybrid CNN models, in which we substitute components of our models across different manifolds. We construct two hybrid model variants: E2H-CNN and H2E-CNN. In E2H-CNN, we use a Euclidean CNN head and a Lorentzian MLR, whereas H2E-CNN employs an HCNN head and a Euclidean MLR. We compare the performance of the two hybrid models to the other three models in Table 8. On TEB datasets, we observe that incorporating a Lorentzian component generally improves performance over a fully Euclidean model, with larger gains from E2H-CNN. These results suggest that using hyperbolic hyperplanes to separate classes may be beneficial, even for Euclidean embeddings. Overall, the results highlight the potential of hybrid models.

Table 8: Model performance (MCC) in TEB, averaged over five random seeds. The best-performing model is bolded.

| Dataset | CNN | HCNN-S | HCNN-M | E2H-CNN | H2E-CNN |
|---|---|---|---|---|---|
| LTR Copia | $54.73_{\pm1.45}$ | $64.58_{\pm3.07}$ | $\mathbf{68.05}_{\pm2.80}$ | $61.82_{\pm2.21}$ | $63.95_{\pm3.52}$ |
| LINEs | $70.63_{\pm1.24}$ | $76.12_{\pm2.16}$ | $77.10_{\pm2.92}$ | $75.65_{\pm0.83}$ | $\mathbf{79.15}_{\pm2.36}$ |
| SINEs | $85.15_{\pm1.64}$ | $85.45_{\pm1.16}$ | $81.85_{\pm2.95}$ | $\mathbf{89.65}_{\pm2.13}$ | $79.49_{\pm3.40}$ |
| CMC-EnSpm | $72.18_{\pm0.32}$ | $\mathbf{80.98}_{\pm1.48}$ | $80.65_{\pm1.30}$ | $76.75_{\pm0.60}$ | $77.15_{\pm3.43}$ |
| hAT-Ac | $87.45_{\pm0.90}$ | $89.61_{\pm1.34}$ | $\mathbf{91.04}_{\pm1.58}$ | $89.76_{\pm0.85}$ | $85.63_{\pm1.44}$ |
| processed | $60.66_{\pm0.82}$ | $\mathbf{68.30}_{\pm0.93}$ | $65.41_{\pm5.54}$ | $66.68_{\pm1.31}$ | $66.12_{\pm0.43}$ |
| unprocessed | $51.94_{\pm2.69}$ | $56.13_{\pm0.56}$ | $\mathbf{58.36}_{\pm1.80}$ | $58.09_{\pm0.96}$ | $58.16_{\pm1.40}$ |

## A.9    $\delta$-HYPERBOLICITY

### A.9.1    ESTIMATION PROCEDURE

Computing $\delta_{\text{worst}}$ naively is an $\mathcal{O}(n^4)$ operation for a set of $n$ points, therefore we use the efficient approach introduced in Khrulkov et al. (2020) and Cohen et al. (2015). Specifically, we incorporate a sampling procedure to estimate hyperbolicity in a computationally tractable manner. The steps are as follows:

1. Sample $N_s$ points from the dataset (we set $N_s = 1000$).
2. Compute the matrix $A$ of pairwise Gromov products using equation 10, and a fixed point $z = z_0$ (detailed in Cohen et al. (2015)).
3. Determine the matrix $C = (A \otimes A) - A$, where $\otimes$ represents the min-max matrix product: $(A \otimes B)_{ij} = \max \min_k \{A_{ik}, B_{kj}\}$.
4. For $\delta_{\text{worst}}$, we take the maximum value from $C$, and for $\delta_{\text{avg}}$, we compute the expected value over the unique elements of $C$ pertaining to valid tuples. We apply the scale-invariant

transformation mentioned in the main text to the $\delta$ values to determine the final values reported. However, for the $\delta_{\mathrm{avg}}$ values, we instead transform the raw values using the scale-invariant ratio introduced in Borassi et al. (2015): $\frac{2\delta_{\mathrm{avg}}}{D_{\mathrm{avg}}}$, where $D_{\mathrm{avg}}$ is the average distance between two randomly selected points.

Results are averaged across multiple runs, and we provide the resulting mean and standard deviation. For the genomic datasets, we use the test set of sequence embeddings generated from the final embedding layer of the trained Euclidean CNN models (Table 9).

### A.9.2 METRIC SPACE CALIBRATIONS

In order to calibrate our $\delta$-hyperbolicity measurements, we scrutinize the behavior of $\delta$ approximations at various fixed curvatures ($K$) and dimensionalities ($d$). We use the MANIFY package, introduced in Chlenski et al. (2025), to randomly sample data points from a Gaussian distribution across different manifolds, using the wrapped normal distribution in the hyperbolic ($K = -1, -2$) (Nagano et al., 2019) and the hyperspherical ($K = 1, 2$) (Skopek et al., 2020) cases. We then compute $\delta$ estimates according to the procedure in A.9.1. We use the geodesic distance of each manifold to determine the distance matrix between points.

The results of the simulations are visualized in Figure 12. The decreasing trend in both $\delta_{\mathrm{worst}}$ and $\delta_{\mathrm{avg}}$ estimates across curvatures suggests that a higher dimensionality of data points may lead to increased hyperbolicity in datasets. For discrete metric spaces, we confirm that for trees, $\delta_{\mathrm{worst}} = \delta_{\mathrm{avg}} = 0$ by using the NETWORKX package (Hagberg et al., 2008) to generate random tree graphs and compute the distance matrix based on shortest paths within each graph.

### A.9.3 DNA LANGUAGE MODELS

We explore the hyperbolicity of sequences embedded by large DNA LMs. Our analysis encompasses a diverse range of pretrained models, selected to represent various architectural approaches and scales. The models under examination are HyenaDNA, DNABERT-2, and NT-500M human.

As a case study, we probe a subset of sequences that likely reflect strongly conserved evolutionary relationships. We therefore generate LM embeddings for a randomly sampled set of SINE sequences from TEB. The embeddings are derived by applying mean pooling over the final layer embedding output of each model. To establish a comparative baseline, we juxtapose the underlying $\delta$ distribution of each LM with a distribution generated from randomly sampled points from a Gaussian of equivalent dimensionality, following the procedure outlined in Section 5.2.

The results of our analysis are presented in Figure 13. Notably, the embeddings produced by HyenaDNA and DNABERT-2 exhibit significantly higher hyperbolicity compared to the baseline embeddings ($p < 0.01$, Wilcoxon rank-sum test). In contrast, the representations generated by the NT-500M human display substantially lower hyperbolicity than the baseline. This disparity may stem from the higher dimensionality of the NT-500M human embeddings, suggesting that hyperbolicity may become less critical at sufficiently large embedding scales.

### A.10 HYPERBOLIC SEQUENCE REPRESENTATIONS

In exploring the sequence representations learned by HCNNs, we build on the intuition introduced by Khrulkov et al. (2020), where hyperbolic image embeddings of MNIST show that ambiguous digits tend to cluster near the center of the Poincaré disk, while clearer, more confidently classified digits lie closer to the boundary. Similarly, in Figure 14, we observe that in the processed pseudogene dataset from TEB, sequence embeddings located near the center of the Poincaré disk (representing the top of the hierarchy) correspond to low-confidence predictions by HCNNs, approximated using model loss. In contrast, embeddings near the boundary exhibit the highest classification confidence. This pattern supports the notion that well-defined sequences occupy lower regions in the hierarchy, where the increased representational capacity of hyperbolic space allows for finer-grained separation based on distinctive sequence features.

To systematically investigate the underlying sequence features informing these hyperbolic genome embeddings, we performed an *in silico* mutagenesis experiment using the processed pseudogene

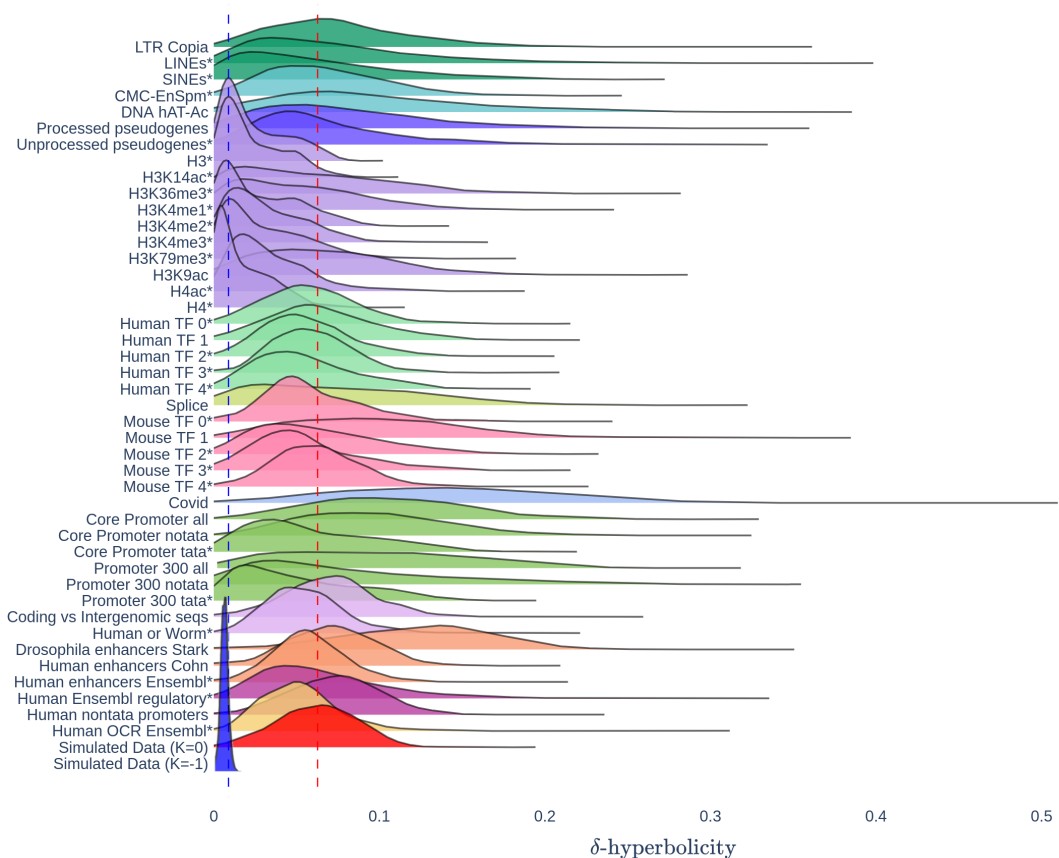

Figure 10: Distribution of scaled $\delta$-hyperbolicity values across each genomic dataset. Colors delineate different task categories, while the bottom two rows provide reference distributions for $\delta$ values computed from a set of points sampled from the normal distribution on a Euclidean ($K = 0$, red) and a hyperbolic ($K = -1$, blue) manifold. Dashed lines indicate the $\delta_{\text{avg}}$ values for the hyperbolic reference (blue) and the Euclidean reference (red). An asterisk (*) denotes that the corresponding distribution constitutes smaller $\delta$ values (i.e., is more hyperbolic) than the Euclidean reference based on the Wilcoxon rank-sum test ($p < 0.01$).

Table 9: $\delta$-Hyperbolicity values of the final embeddings for CNNs trained on each genomic dataset. Results are averaged over 10 sampling runs (mean $\pm$ standard deviation).

| Benchmark | Task | Dataset | $\delta_{\text{worst}}$ | $\delta_{\text{avg}}$ |
|---|---|---|---|---|
| TEB | Retrotransposons | LTR Copia | $0.36_{\pm 0.0175}$ | $0.145_{\pm 0.0019}$ |
| | | LINEs | $0.40_{\pm 0.0110}$ | $0.164_{\pm 0.0004}$ |
| | | SINEs | $0.08_{\pm 0.0076}$ | $0.170_{\pm 0.0016}$ |
| | DNA transposons | CMC-EnSpm | $0.18_{\pm 0.0181}$ | $0.163_{\pm 0.0009}$ |
| | | hAT-Ac | $0.37_{\pm 0.0220}$ | $0.215_{\pm 0.0026}$ |
| | Pseudogenes | processed | $0.36_{\pm 0.0204}$ | $0.189_{\pm 0.0007}$ |
| | | unprocessed | $0.35_{\pm 0.0140}$ | $0.157_{\pm 0.0003}$ |
| GUE | Epigenetic Marks Prediction | H3 | $0.10_{\pm 0.0072}$ | $0.098_{\pm 0.0005}$ |
| | | H3K14ac | $0.09_{\pm 0.0090}$ | $0.101_{\pm 0.0030}$ |
| | | H3K36me3 | $0.26_{\pm 0.0541}$ | $0.251_{\pm 0.0014}$ |
| | | H3K4me1 | $0.21_{\pm 0.0185}$ | $0.225_{\pm 0.0056}$ |
| | | H3K4me2 | $0.13_{\pm 0.0112}$ | $0.125_{\pm 0.0039}$ |
| | | H3K4me3 | $0.14_{\pm 0.0168}$ | $0.169_{\pm 0.0020}$ |
| | | H3K79me3 | $0.15_{\pm 0.0255}$ | $0.122_{\pm 0.0067}$ |
| | | H3K9ac | $0.21_{\pm 0.0160}$ | $0.265_{\pm 0.0058}$ |
| | | H4ac | $0.18_{\pm 0.0156}$ | $0.186_{\pm 0.0024}$ |
| | | H4 | $0.10_{\pm 0.0058}$ | $0.082_{\pm 0.0041}$ |
| | Human Transcription Factor Prediction | 0 | $0.20_{\pm 0.0114}$ | $0.160_{\pm 0.0026}$ |
| | | 1 | $0.20_{\pm 0.0245}$ | $0.152_{\pm 0.0044}$ |
| | | 2 | $0.19_{\pm 0.0189}$ | $0.148_{\pm 0.0021}$ |
| | | 3 | $0.19_{\pm 0.0189}$ | $0.141_{\pm 0.0004}$ |
| | | 4 | $0.18_{\pm 0.0098}$ | $0.140_{\pm 0.0009}$ |
| | Splice Site Prediction | splice | $0.29_{\pm 0.0363}$ | $0.256_{\pm 0.0012}$ |
| | Mouse Transcription Factor Prediction | 0 | $0.21_{\pm 0.0147}$ | $0.140_{\pm 0.0043}$ |
| | | 1 | $0.35_{\pm 0.0301}$ | $0.249_{\pm 0.0032}$ |
| | | 2 | $0.21_{\pm 0.0226}$ | $0.139_{\pm 0.0011}$ |
| | | 3 | $0.19_{\pm 0.0237}$ | $0.131_{\pm 0.0009}$ |
| | | 4 | $0.19_{\pm 0.0112}$ | $0.148_{\pm 0.0022}$ |
| | Covid Variant Classification | covid | $0.50_{\pm 0.0388}$ | $0.417_{\pm 0.0030}$ |
| | Core Promoter Detection | all | $0.29_{\pm 0.0105}$ | $0.229_{\pm 0.0034}$ |
| | | notata | $0.28_{\pm 0.0184}$ | $0.212_{\pm 0.0010}$ |
| | | tata | $0.22_{\pm 0.0082}$ | $0.138_{\pm 0.0013}$ |
| | Promoter Detection | all | $0.29_{\pm 0.0146}$ | $0.260_{\pm 0.0024}$ |
| | | notata | $0.31_{\pm 0.0210}$ | $0.257_{\pm 0.0043}$ |
| | | tata | $0.16_{\pm 0.0127}$ | $0.138_{\pm 0.0069}$ |
| GB | Demo | coding vs intergenomic seqs | $0.21_{\pm 0.0180}$ | $0.118_{\pm 0.0019}$ |
| | | human or worm | $0.19_{\pm 0.0189}$ | $0.121_{\pm 0.0010}$ |
| | Enhancers | drosophila enhancers stark | $0.30_{\pm 0.0174}$ | $0.209_{\pm 0.0012}$ |
| | | human enhancers cohn | $0.19_{\pm 0.0137}$ | $0.092_{\pm 0.0002}$ |
| | | human enhancers ensembl | $0.19_{\pm 0.0198}$ | $0.109_{\pm 0.0001}$ |
| | Regulatory | human ensembl regulatory | $0.23_{\pm 0.0282}$ | $0.148_{\pm 0.0013}$ |
| | | human non-tata promoters | $0.19_{\pm 0.0053}$ | $0.103_{\pm 0.0002}$ |
| | Open Chromatin Regions | human ocr ensembl | $0.24_{\pm 0.0400}$ | $0.189_{\pm 0.0011}$ |

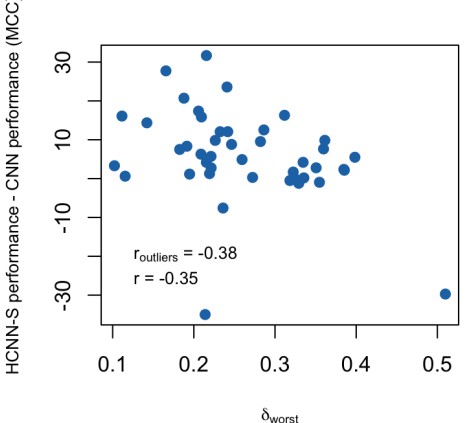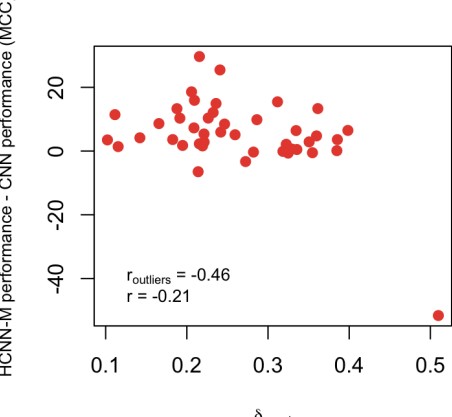

Figure 11: Correlation between $\delta_{\text{worst}}$ and the performance differential between HCCN-S and CNN models. $r_{\text{outliers}}$ includes outliers in the Pearson correlation coefficient calculation, while $r$ excludes them ($p < 0.05$, except for HCNN-M $r$).

dataset. Our methodology involves a structured approach to dissecting sequence representations. For a given sequence, we:

1. Retrieve the Genomic Evolutionary Rate Profiling (GERP) Cooper et al. (2005) score for each nucleotide along the sequence. GERP scores quantify evolutionary constraints at specific genomic positions, identifying which positions are functionally important based on selective pressure. GERP uses multiple sequence alignments across species to identify conserved regions.

2. Strategically mutate a subset of nucleotides under the highest selective pressure, generating multiple perturbed sequence variants.

3. Generate embeddings for each perturbed sequence using the trained HCNN.

Figure 15 visualizes this experiment using both a processed pseudogene sequence and a background sequence. As mutations progressively erode the evolutionary signal associated with strong selection (corresponding to the nucleotides with the highest GERP scores), the features rendering the pseudogene "gene-like" may deteriorate. This degradation increases sequence ambiguity from the HCNN's perspective, manifesting as a shift of the perturbed representations toward the top of the hierarchy—near the center of the Poincaré disk—where low-confidence sequences typically reside. The loss of these evolutionary features actively hinders the model's ability to recognize pseudogenes.

Critically, this effect is sequence-specific: perturbing conserved regions within noisy background sequences fails to produce an equivalent shift, suggesting the model prioritizes features consistently associated with the pseudogene class.

To validate the generalizability of this phenomenon, we conducted a comprehensive analysis on a randomly sampled set of 10,000 sequences from the processed pseudogene dataset, ensuring balanced class representation. For each sequence, we applied our mutagenesis protocol (steps 1-3), generating 10 perturbed sequence variants and corresponding HCNN representations.

We quantify this effect by measuring the embedding shifts of these perturbed sequences relative to their original embeddings, specifically tracking their movement toward the representation space's origin. This directional analysis provides insight into the HCNN's representational sensitivity: a trajectory toward the origin likely indicates increased representational ambiguity for the perturbed sequence. The distance between each sequence (perturbed and original) and the origin in the embedding space is computed using Poincaré geodesics.

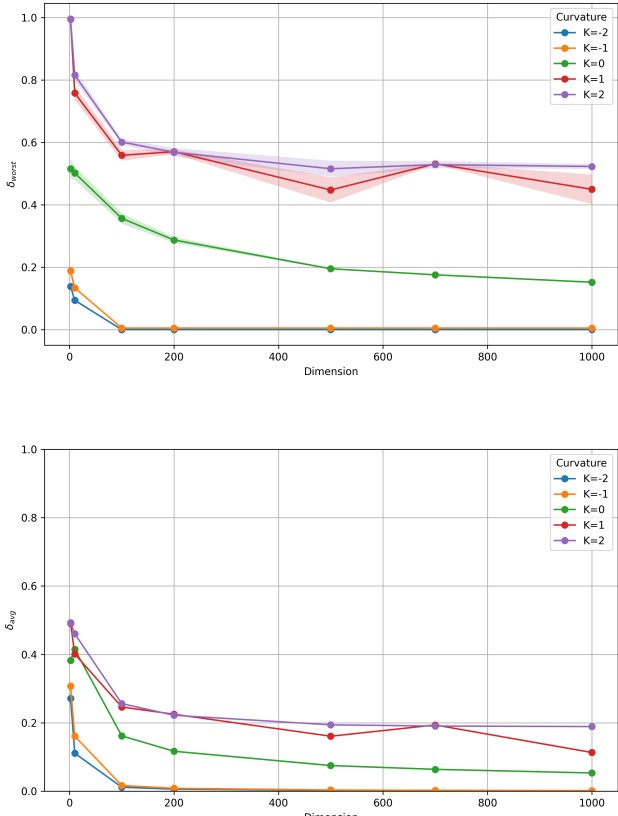

Figure 12: Estimates for $\delta_{\mathrm{worst}}$ (top) and $\delta_{\mathrm{avg}}$ (bottom) using simulated data points from a wrapped normal distribution on manifolds with varying curvatures ($K$) and dimensionalities.

When comparing representational shifts between perturbed processed pseudogene and background sequences, we observe significantly greater movement toward the Poincaré disk's origin for perturbed pseudogene sequences. This difference, statistically validated by the Wilcoxon rank-sum test ($p < 0.05$), demonstrates the robustness of our findings. We tested a range of mutation rates, altering $10 - 30\%$ of nucleotides per sequence, and found that the effect remained consistent and statistically significant across all rates.

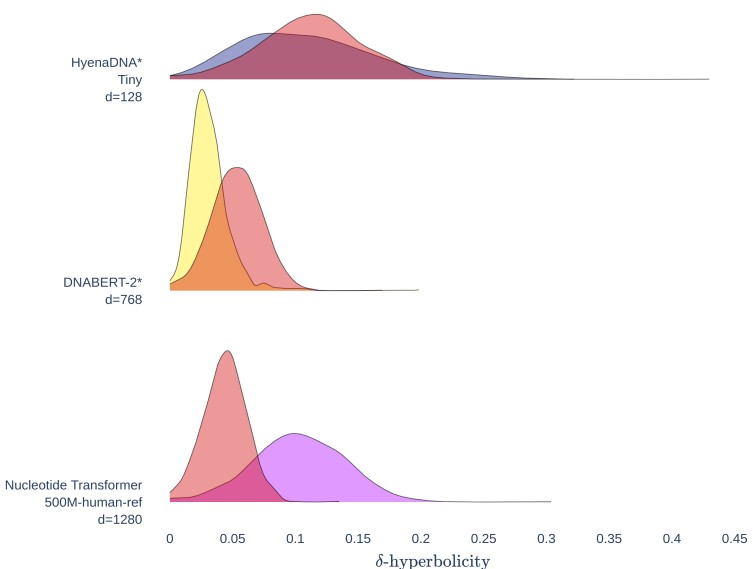

Figure 13: Distribution of scaled $\delta$-hyperbolicity values using embeddings from various DNA LMs. The distribution of each model is overlaid with the $\delta$ distribution of randomly sampled points from a Gaussian of equal dimensionality (red). An asterisk (*) denotes that the corresponding distribution has significantly smaller smaller $\delta$ values (i.e., is more hyperbolic) than the Euclidean reference, based on the Wilcoxon rank-sum test ($p < 0.01$).

Table 10: Mean model performance (MCC) aggregated by genomics task (mean $\pm$ standard error).

| Benchmark | Task | Euclidean CNN | Hyperbolic HCNN-S | Hyperbolic HCNN-M |
|---|---|---|---|---|
| | | | Model | |
| TEB | Retrotransposon Prediction | $70.48_{\pm8.02}$ | $74.80_{\pm13.48}$ | $\mathbf{75.98}_{\pm12.48}$ |
| | DNA transposon Prediction | $79.91_{\pm10.85}$ | $85.30_{\pm13.82}$ | $\mathbf{85.77}_{\pm20.37}$ |
| | Pseudogene Prediction | $56.30_{\pm7.40}$ | $\mathbf{62.22}_{\pm10.26}$ | $60.31_{\pm11.66}$ |
| GUE | Epigenetic Marks Prediction | $40.76_{\pm4.07}$ | $\mathbf{55.31}_{\pm2.64}$ | $48.18_{\pm2.81}$ |
| | Human Transcription Factor Prediction | $52.52_{\pm3.63}$ | $61.12_{\pm3.12}$ | $\mathbf{61.25}_{\pm2.86}$ |
| | Splice Site Prediction | $78.64_{\pm0.19}$ | $80.32_{\pm0.55}$ | $\mathbf{80.76}_{\pm0.47}$ |
| | Mouse Transcription Factor Prediction | $45.79_{\pm4.72}$ | $\mathbf{61.93}_{\pm5.52}$ | $61.52_{\pm5.08}$ |
| | Core Promoter Detection | $70.13_{\pm2.06}$ | $70.12_{\pm3.48}$ | $\mathbf{70.99}_{\pm2.39}$ |
| | Promoter Detection | $\mathbf{85.80}_{\pm1.75}$ | $85.73_{\pm1.37}$ | $85.66_{\pm1.82}$ |
| | Covid Variant Classification | $\mathbf{66.43}_{\pm0.21}$ | $36.71_{\pm4.33}$ | $14.81_{\pm0.21}$ |
| GB | Demo | $82.52_{\pm2.79}$ | $86.34_{\pm2.83}$ | $\mathbf{86.48}_{\pm2.83}$ |
| | Enhancers | $\mathbf{39.41}_{\pm9.00}$ | $34.18_{\pm7.46}$ | $28.77_{\pm2.83}$ |
| | Regulatory | $77.36_{\pm4.73}$ | $\mathbf{86.74}_{\pm1.35}$ | $85.05_{\pm2.34}$ |
| | Open Chromatin Regions | $39.92_{\pm0.38}$ | $\mathbf{56.22}_{\pm0.13}$ | $55.36_{\pm1.23}$ |

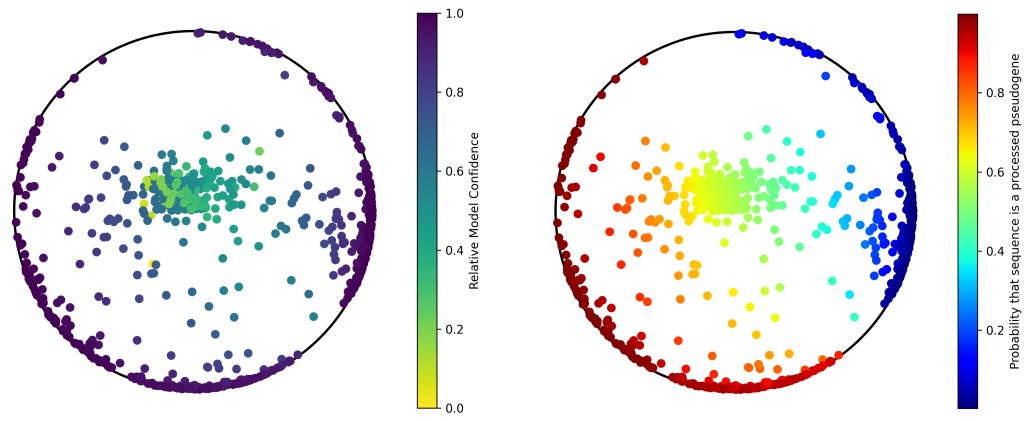

Figure 14: HCNN embeddings for the processed pseudogene dataset, colored by model confidence on the left and by the probability that a sequence is a processed pseudogene (vs. a background sequence) on the right. Sequence embeddings are visualized on the Poincaré disk.

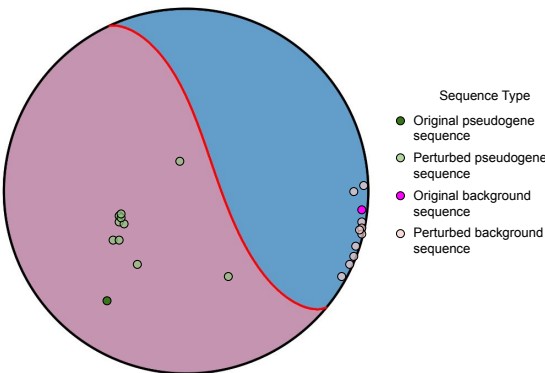

Figure 15: HCNN embeddings for a processed pseudogene sequence and a background sequence. Each sequence has been perturbed multiple times, with different instances shown on the Poincaré disk.

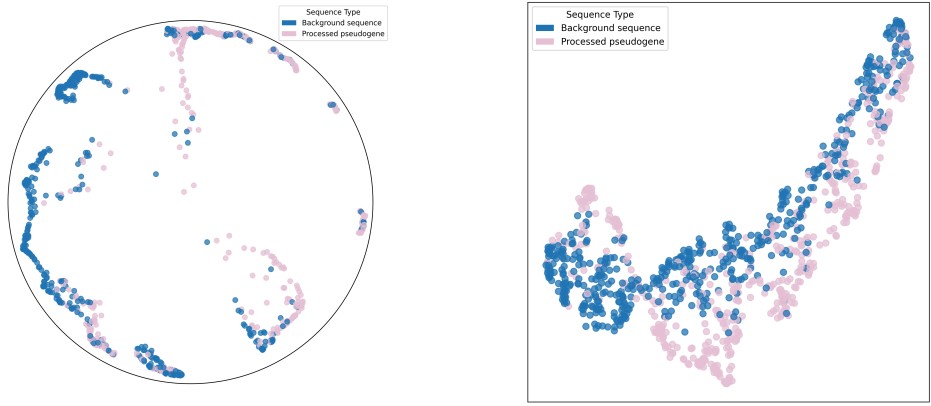

Figure 16: UMAP visualizations of the embeddings generated by the HCNN (left) and CNN (right) trained on the processed pseudogene dataset in TEB.

