# OpenReview forum: "Hyperbolic Genome Embeddings"
_ICLR.cc/2025/Conference — ICLR 2025 Poster_

### Official Review · Reviewer_KNja · 2024-10-28

**Soundness:** 2
**Presentation:** 3
**Contribution:** 3
**Rating:** 8
**Confidence:** 4

**Summary:**

The paper proposes the application of fully hyperbolic CNNs (HCNNs) for representation of biological data, in specific DNA sequences. The authors note that genomic data is often generated by evolutionary processes, and result in phylogenetic hierarchies that may be better represented in hyperbolic space as opposed to Euclidean space. The authors apply HCNNs on several genomic element classification benchmarks, and additionally propose a new dataset, the Transposable Elements Benchmark, in which models are tasked with classifying the type of transposable element that exists in a context window. The authors find that hyperbolic CNNs outperform their Euclidean counterparts in most of the benchmark tasks, and can occasionally outperform genomic foundation model performance. They also perform an empirical analysis on the hyperbolicity of the selected datasets.

NOTE: Score raised from 5 -> 8 in response to rebuttal.

**Strengths:**

- Paper presents a creative application of HCNNs to the field of genomics, and serves as a useful preliminary paper to explore the benefits of representing genomic data in hyperbolic space. Given the current community interest in representation learning for genomics, this new approach could be of high interest and significance. Authors also provide new dataset for benchmarking a relevant task.

- The authors make an effort to construct high quality baselines for their experiments. In addition to demonstrating the benefits of an HCNN vs a Euclidean CNN, the method is also benchmarked against genomic foundation models. The authors also take care during their empirical evaluation of dataset hyperbolicity to account for dataset dimensionality by calibrating against simulated data.

- Experimental results indicate the hyperbolic representation offers benefit on some classes of tasks. This shows that the HCNN or other hyperbolic representations could have promise in genomic representation learning. Visually, it appears that the tasks where HCNNs offer improvement also correlate with the underlying data hyperbolicity, supporting the core hypothesis of the work.

- Experiments are well executed, with replication across random seeds and confidence intervals + statistical significance reported. Experimental setup is well described, and code is available, indicating that results are likely to be reproducible.

**Weaknesses:**

- I believe the analysis could be improved by investigating exactly how the HCNN represents sequences related through evolution. A common intuition I have seen for hyperbolic embeddings in computer vision is that datum closer to the origin tend to represent high level features and vice versa. It would be convincing if some analogy could be made for the genomic sequences through the experiments. For example, while Figure 2 demonstrates that hierarchical structure exists in the HCNN representation, it would be interesting to see whether the sequences placed at the "top" of the learned hierarchy reconcile with biological intuition.

- It's surprising that the HCNN representation is not consistently better for datasets that seem highly related to evolutionary processes. For example, it is strange that the covid variant classification is so poor given that the observed variation is likely driven by selection, and that the HCNN-M is outperformed by the Euclidean CNN on the synthetic dataset which is generated using a evolutionary simulator. Similarly, it is interesting that the HCNN does not significantly outperform the Euclidean models on promoter classification tasks despite the results in the empirical study in Figure 3. I would appreciate any deeper insights from the authors on these issues.

- Tying into the last point, given the lack of consistent improvement and correlation in improvement with seemingly evolutionary data generating processes, its unclear under what circumstance the HCNN should be used compared to existing approaches.

- To my knowledge, there is very limited methodological novelty in the paper, although I believe the current paper may be sufficient for an applied paper.

**Questions:**

- I would appreciate if the authors could address the above points.

- Could selection of GUE as a benchmark cause underperformance? From my understanding, the benchmark does not control for homology in the train / test splits. Conceivably, the benefit of the HCNN would be improved generalization in evolutionary / phylogenetic contexts, and potential data leakage in the GUE benchmark could mask the performance benefit in this aspect.

- I would appreciate if the colour of the DNABERT-2 distribution in supplemental figure 12 could be changed to be more distinct for folks with colour vision deficiency!

---

> ### Author Response · Authors · 2024-11-22
> **Response to Reviewer KNja (part 1)**
>
> We thank the reviewer for this positive assessment of our work. We appreciate your recognition of the paper's many strengths, including its significant impact on the genomics research community, the rigorous and high-quality execution of experiments, and the comprehensive benchmarking efforts. We are also grateful for the thoughtful questions you raised which prompted discerning experiments that have strengthened the paper and unified the motivating arguments.
>
> > I believe the analysis could be improved by investigating exactly how the HCNN represents sequences related through evolution. A common intuition I have seen for hyperbolic embeddings in computer vision is that datum closer to the origin tend to represent high level features and vice versa. It would be convincing if some analogy could be made for the genomic sequences through the experiments. For example, while Figure 2 demonstrates that hierarchical structure exists in the HCNN representation, it would be interesting to see whether the sequences placed at the "top" of the learned hierarchy reconcile with biological intuition.
>
> Thanks for bringing up an interesting consideration. Overall, we report two findings:
> 1. Sequences embeddings near the top of the hierarchical structure represent ambiguous sequences (visualized in Figure 15 of the updated manuscript).
> 2. Ablating conserved regions of sequence (corresponding to regions with strong evolutionary signal) results in the loss of definitive features for sequences, shifting them closer to the top (more ambiguous part) of the hierarchy (visualized in Figure 16 of the updated manuscript). Thus, sequences with definitive features with respect to the class label lie closer to the Poincaré disk boundary.
>
> In exploring the sequence representations of HCNNs, we started with the intuition built in [1], where hyperbolic image embeddings of MNIST (Figure 1 of [1]) near the center of the Poincaré  disk represent the most ambiguous looking digits, while clear images lie near the boundary. Similarly, in the new Figure 15, we observe that in the processed pseudogene dataset in TEB, the sequence embeddings that lie close to the center of the Poincaré disk (the top of the hierarchy) correspond to low confidence embeddings for HCNNs, while the embeddings near the disk boundaries show the highest classification confidence. This is consistent with the idea that well defined sequences are at the bottom of the hierarchy where there is more space to separate out nuanced differences between sequences based on distinctive features.
>
> Next, since we cannot readily identify features from DNA sequence as one can with images, we conduct an experiment to dissect the sequence features informing the hyperbolic genome embedding. Given our dataset of processed pseudogenes, we examine the changes made to the HCNN representation by perturbing a fixed pseudogene sequence. For a fixed sequence, we:
>
> 1. Compute the Genomic Evolutionary Rate Profiling (GERP) [2] score for each nucleotide along the sequence. GERP scores quantify evolutionary constraints at specific genomic positions, identifying which positions are functionally important based on selective pressure.
> 2. Mutate a fraction of the nucleotides under the highest selective pressure.
> 3. Use our HCNN to generate an embedding for this perturbed instance of our original sequence.
>
> Figure 16 visualizes this experiment using a processed pseudogene sequence and a background sequence. As the evolutionary signal under strong selection is eroded by the introduced mutations, it is likely that the features that make the pseudogene more “gene-like” are degraded. This degradation ultimately makes the sequence more ambiguous to the HCNN, and we observe that the perturbed representations move closer to the top of the hierarchy (near the center of the Poincaré disk), where the low confidence sequences lie. Removal of these evolutionary features actively hinders the model in identifying pseudogenes. However, perturbing conserved regions from the noisy background sequences does not appear to have this effect, as the model focuses on learning features common to the pseudogene class. Please see section A.10 of the updated paper for visualizations and writeup.
>
> [1] Khrulkov, V., Mirvakhabova, L., Ustinova, E., Oseledets, I., & Lempitsky, V. (2020). Hyperbolic image embeddings. In Proceedings of the IEEE/CVF conference on computer vision and pattern recognition (pp. 6418-6428).
>
> [2] Cooper, G. M., Stone, E. A., Asimenos, G., Green, E. D., Batzoglou, S., & Sidow, A. (2005). Distribution and intensity of constraint in mammalian genomic sequence. Genome research, 15(7), 901-913.

---

> > ### Author Response · Authors · 2024-11-22
> > **Response to Reviewer KNja (part 2)**
> >
> > > It's surprising that the HCNN representation is not consistently better for datasets that seem highly related to evolutionary processes. For example, it is strange that the covid variant classification is so poor given that the observed variation is likely driven by selection, and that the HCNN-M is outperformed by the Euclidean CNN on the synthetic dataset which is generated using a evolutionary simulator.  I would appreciate any deeper insights from the authors on these issues.
> >
> > Thank you for the perceptive comment. We, too, found the covid variant classification and synthetic dataset results surprising. We sought to further explain these observations by reconsidering the various plausible data-generating processes for biological sequences. We defined three potential cases of biological signal transmission that might be learned by the models, and constructed 3 classes of synthetic datasets:
> >
> > (A) Intra-tree differentiation: this was our original synthetic experiment. In this scenario, we generated sequences from a single phylogenetic tree, with labels derived from clade membership. This experiment tested whether a hyperbolic model would better differentiate parts of a single phylogenetic tree from each other.
> >
> > (B) Inter-tree differentiation: in this scenario sequences are generated from different phylogenetic trees, with labels derived from phylogeny membership. This experiment tests whether the model is differentiating different evolutionary processes from one another.
> >
> > (C) Tree identification: In this scenario, sequences for one label are generated from a phylogenetic tree, whereas the other label identifies sequences that are either completely randomly generated or sampled from background sequence (and therefore without any guiding evolutionary relationship). This experiment tests whether the model is differentiating evolutionary processes from noise.
> >
> > Scenarios are visualized in the new Figure 2 in the paper, and detailed in section 4.1 - Synthetic Datasets.
> >
> > We additionally tested two variations of synthetic datasets - one in which sequences were completely randomly generated, and one in which sequences are sampled from existing genomes to see if the fully artificial sequences we originally used were not generalizable to our real world dataset results. Overall, we constructed 5 additional experiments, which we share the results of in the new Table 2 in the paper and reproduced here (with full experimental details in A.6):
> >
> > | **Scenario** | **Sequence**  | **Euclidean CNN**      | **Hyperbolic HCNN-S**        | **Hyperbolic HCNN-M**        |
> > |--------------|---------------|-------------------------|------------------------------|------------------------------|
> > | **A**        | Artificial    | 62.38 ± 2.28           | **65.25 ± 3.27**             | 59.25 ± 2.60                |
> > |              | Real          | 61.72 ± 3.08           | **66.44 ± 3.14**             | 61.26 ± 2.99                |
> > | **B**        | Artificial    | 58.50 ± 0.82           | **60.53 ± 0.80**             | 59.75 ± 0.54                |
> > |              | Real          | 57.50 ± 0.88           | **62.53 ± 6.94**             | 59.12 ± 0.54                |
> > | **C**        | Artificial    | 62.05 ± 1.62           | **67.65 ± 1.09** †           | 67.43 ± 1.57 †              |
> > |              | Real          | 66.22 ± 0.44           | **73.62 ± 0.62** †           | 69.30 ± 2.34 †              |
> >
> > We are pleased to report that the results of these experiments provide a clearer understanding of how the hyperbolic inductive bias functions in a learning context. In the genomics domain, **this bias primarily operates by uncovering the underlying phylogenetic tree structure amidst noise.** While this learning mechanism may also offer advantages in disentangling distinct evolutionary patterns (Scenario B), the results consistently indicate that excelling in Scenario C is somewhat orthogonal to excelling in Scenario A. This is likely because sequence differentiation is performed at the tree level rather than at the clade level. Notably, this finding aligns with results observed on real-world datasets: the most significant performance gains are observed in scenarios where an evolutionary signal (e.g., transcription factor binding sites, epigenetic marks, or transposable elements) is effectively distinguished from background noise (e.g., background sequences).
> >
> > Overall, we find these results to support our original hypothesis discussed in the introduction of our paper - the inductive biases of our models extend past approximating explicit edit distance mutations to instead focus on latent hierarchical structure that likely occupies a lower dimensional manifold.

---

> > > ### Author Response · Authors · 2024-11-22
> > > **Response to Reviewer KNja (part 3)**
> > >
> > > > For example, it is strange that the covid variant classification is so poor given that the observed variation is likely driven by selection, and that the HCNN-M is outperformed by the Euclidean CNN on the synthetic dataset which is generated using a evolutionary simulator.
> > >
> > > Given the findings from the previous comment, we now have an understanding for why HCNNs are comparatively disadvantaged in the covid variant classification task. The covid dataset is designed by sampling random portions of each covid variant genome. The underlying sequence-generating process behind covid variant evolution most closely resembles Scenario A in our synthetic dataset experiments - in which a single ancestral sequence evolves along multiple paths. Since the task is designed to differentiate sequences across covid variants, it is similar to differentiating separate clades on a tree. As we concluded earlier, the inductive biases of our model are better suited to identifying phylogenetic signal amidst noise than in differentiating within the phylogenetic tree. Therefore, the lower performance on the covid variant task is consistent with our understanding of how HCNNs learn biological signal.
> > >
> > > > Similarly, it is interesting that the HCNN does not significantly outperform the Euclidean models on promoter classification tasks despite the results in the empirical study in Figure 3. I would appreciate any deeper insights from the authors on these issues.
> > >
> > > From Figure 3 (Figure 11 in the updated manuscript) we can observe that, with the exception of the tata only datasets (where we reach SOTA performance), the promoter data is not significantly hyperbolic and therefore may not be well-suited for HCNNs. As promoters likely function through highly complex, often combinatorial interactions, we hypothesize that these latent elements may be more challenging for HCNNs to capture.
> > >
> > > > Tying into the last point, given the lack of consistent improvement and correlation in improvement with seemingly evolutionary data generating processes, its unclear under what circumstance the HCNN should be used compared to existing approaches.
> > >
> > > Thanks to your suggestions, we have performed careful experiments that identify a much more consistent learning pattern for HCNNs. We can conclude that HCNNs are beneficial in the set of use cases where we want to discern underlying evolutionary signal from noise. These types of use cases are highly prevalent in genomics research and encompass a wide variety of biological processes including transcription factor binding, epigenetic marks, transposable elements, splice sites, and chromatin regions. Indeed, these areas have encompassed the vast majority of research applications for representation learning in genomics.
> > >
> > > > To my knowledge, there is very limited methodological novelty in the paper, although I believe the current paper may be sufficient for an applied paper.
> > >
> > > While we primarily consider our work to be an applications paper, we have also created a novel approach to evaluating the hyperbolicity of datasets, by considering the distributional shifts that occur in hyperbolicity both with respect to the underlying data generating processes and data dimensionality. We show that Gromov hyperbolicity correlates significantly with improvements in performance from a hyperbolic model, providing further evidence that hyperbolic models are well-suited to genomics.

---

> > > > ### Author Response · Authors · 2024-11-22
> > > > **Response to Reviewer KNja (part 4)**
> > > >
> > > > > Could selection of GUE as a benchmark cause underperformance? From my understanding, the benchmark does not control for homology in the train / test splits. Conceivably, the benefit of the HCNN would be improved generalization in evolutionary / phylogenetic contexts, and potential data leakage in the GUE benchmark could mask the performance benefit in this aspect.
> > > >
> > > > Thanks for raising a pertinent question. We performed an experiment to determine whether the inductive biases of HCNNs improve generalizability to unseen homology branches. In order to validate this hypothesis, we assess the zero-shot capability of our model in identifying sequences originating from an unseen phylogenetic tree against random background sequences. For our training data, we generate a synthetic dataset as we did for testing Scenario C (sequences generated from the tree share the same label, and sequences not from the tree share a different label). However, instead of splitting this dataset into train/validation/test sequences, we create our test set separately by generating a completely new phylogenetic tree and sampling sequences from this set. The tree-generated sequences in the test dataset thus originate from completely unseen homology branches. This process is visualized in the new Figure 9 and further detailed in section A.7 of the paper.
> > > >
> > > > | CNN | HCNN-S | HCNN-M |
> > > > | ------- |------- | ------- |
> > > > | 24.31 &pm; 7.99 | **45.73 &pm; 8.93** † | 40.87 &pm; 8.93 † |
> > > >
> > > > †Significant with p < 0.05, Wilcoxon rank-sum test
> > > >
> > > > From our results, we observe that the hyperbolic models gain a significant advantage over the Euclidean model, which validates your hypothesis. This finding would suggest that the inductive biases of a hyperbolic model offer an even larger advantage over Euclidean models than originally estimated, since most genomic datasets do not account for this effect and may therefore overestimate performance of prediction methods.
> > > >
> > > > > I would appreciate if the colour of the DNABERT-2 distribution in supplemental figure 12 could be changed to be more distinct for folks with colour vision deficiency!
> > > >
> > > > Thank you for bringing our attention to this issue. We have updated the color in Figure 12 (Figure 14 in the updated manuscript) to be more visually distinct for individuals with color vision deficiency.

---

> ### Comment · Reviewer_KNja · 2024-11-22
>
> Thank you for conducting a thorough rebuttal, it was genuinely exciting to see the results of the additional analysis. I found the perturbation experiments particularly creative and compelling. I believe the set of new experiments successfully identifies the hypothetical use-cases and representational advantages of the HCNN in genomics, and consequently will be of high practical value to the community. I will raise my score to reflect the improved manuscript.

---

### Official Review · Reviewer_H56n · 2024-10-30

**Soundness:** 2
**Presentation:** 2
**Contribution:** 3
**Rating:** 8
**Confidence:** 4

**Summary:**

The authors propose an architecture that is better suited for genomic property prediction from DNA sequences. They implement hyperbolic convolutional neural networks (HCNN) and demonstrate improvements across a number of tasks. They compare against a matched CNN in Euclidean space. They motivate the hyperbolic embeddings due to the tree like underlying generative process of genomic sequences. In addition the authors propose an additional Transposable Elements Benchmark. In addition they use a measure of $\delta$ hyperbolicity or a measure of tree-likeness in genetic data.

**Strengths:**

- The authors analyze an existing problem through a novel lens suggesting a new possible inductive bias to an important problem.
- The authors perform a comprehensive evaluation across a wide variety of tasks
- Authors propose a new dataset which can evaluate ability of models to classify transposable elements. This class of sequences are widely abundant however it is unclear as to their biological significance compared to sequences encoding mature RNAs.

**Weaknesses:**

- This work can benefit from additional  justification for why the hierarchical tree structure is a good inductive bias for genomic data. Although the original data generating process can take on a tree structure is it necessarily the case that we would want to add this as an inductive bias to the model?
- Is there an experiment that the authors can do that would confirm the validity of this inductive bias? Do the authors find it strange that the synthetic task that was explicitly designed to test performance of the inductive bias the baseline falls within the confidence interval of the best performing model?
- In genomic sequence modeling it is common to do homology splitting where sequences related through their homologous relationships are fully left out and the model's capacity is assessed on whether it's able to generalize to unseen homology branches. Do the authors expect that this inductive bias helps or hurts the performance of the model in this domain? Can they perform an experiment validating this?

**Questions:**

- If the hierarchical relationship is between sequences then why do the authors note that a single point would be an individual nucleotide and a sequence would be a set? Isn't a sequence then exponential in the number of individual nucleotides thus resulting in an exponential growth of possibility space even without the hyperbolic space inductive bias?
- There is no mention regarding the parameter count of the eucledian CNN. I assume it is the same as the hyperbolic version but would be good to get a confirmation. Can the authors also include FLOP count per forward / backward pass for each of the models?
- The work can benefit from an expanded more thorough explanation in the appendix of mapping between tangent space and Lorentz manifold especially when describing the hypebolic layers.
- Can the authors mention how the confidence intervals were generated in table 1.
- Do the authors find it strange that for covid variant classification, arguably in a very clear cut case where there is abundant information regarding evolution of the virus the hyperbolic architecture doesn't perform so well? Do the authors have a hypothesis for why that is the case?

---

> ### Author Response · Authors · 2024-11-19
> **Response to Reviewer H56n (part 1)**
>
> Thank you for taking the time to write your thoughtful review. We are glad to see that you recognize the novelty of our approach and the comprehensiveness of our evaluation. We are very appreciative of the discerning comments you have provided, and believe that the questions you have raised have led to illuminating experiments that have strengthened the paper and unified the motivating arguments. We provide specifics in our inline responses.
>
> > Authors propose a new dataset which can evaluate ability of models to classify transposable elements. This class of sequences are widely abundant however it is unclear as to their biological significance compared to sequences encoding mature RNAs.
>
> Thanks for raising this point - we realize that we did not provide adequate background on the biological significance of transposable elements (TEs). We have clarified this in the paper by adding the following explanation (with citations):
>
> TEs can influence gene expression and regulation by acting as alternative promoters, providing transcription factor binding sites, introducing alternative splicing, mediating epigenetic modifications, and being co-opted as regulatory elements throughout evolution. As such, TEs have been also implicated in disease pathogenesis.
>
> > W1: This work can benefit from additional justification for why the hierarchical tree structure is a good inductive bias for genomic data. Although the original data generating process can take on a tree structure is it necessarily the case that we would want to add this as an inductive bias to the model?
>
> Ultimately, most of the inductive biases in deep learning genomics models have been inherited from the language and image deep learning domains. While some of these inductive biases are applicable to genomics, many lack strong theoretical or empirical justification and persist primarily due to the absence of alternative frameworks. Our work functions as a challenge to the status quo reliance on Euclidean models in the field of genomics.
>
> Given that evolution is the guiding principle behind all sequence organization  [1, 2], incorporating an inductive bias that mirrors evolutionary processes is essential. We demonstrate this necessity through two key approaches:
>
> 1. Empirical Evidence: We show strong, statistically significant performance improvements across a wide variety of genomics problems pertinent to the field.
>
> 2. Theoretical Framework: We reformulate a well-established measure of hyperbolicity to illustrate the alignment between our approach and the unique, inherent characteristics of genomic data.
>
> [1] Shapiro, J. A. (2002). Genome organization and reorganization in evolution: Formatting for computation and function. Annals of the New York Academy of Sciences, 981(1), 111–134. https://doi.org/10.1111/j.1749-6632.2002.tb04917.x
>
> [2] Lynch, M., & Conery, J. S. (2006). The origins of eukaryotic gene structure. Molecular Biology and Evolution, 23(2), 450–468. https://doi.org/10.1093/molbev/msj050

---

> > ### Author Response · Authors · 2024-11-19
> > **Response to Reviewer H56n (part 2)**
> >
> > > W2: Is there an experiment that the authors can do that would confirm the validity of this inductive bias? Do the authors find it strange that the synthetic task that was explicitly designed to test performance of the inductive bias the baseline falls within the confidence interval of the best performing model?
> >
> > Thank you for raising an insightful question. We, too, found it surprising that the synthetic task did not provide a compelling explanation for why the inductive biases of a hyperbolic model would enhance genome sequence modeling. Given the inconclusive nature of these results, we further considered the various plausible data-generating processes for biological sequences. Ultimately, we defined three potential cases of biological signal transmission that might be learned by the models (which we visualize in the new Figure 2 in the paper, and detail in section 4.1 - Synthetic Datasets):
> >
> > A. Intra-tree differentiation: this was our original synthetic experiment. In this scenario, we generated sequences from a single phylogenetic tree, with labels derived from clade membership. This experiment tested whether a hyperbolic model would better differentiate parts of a single phylogenetic tree from each other.
> >
> > B. Inter-tree differentiation: in this scenario sequences are generated from different phylogenetic trees, with labels derived from phylogeny membership. This experiment tests whether the model is differentiating different evolutionary processes from one another.
> >
> > C. Tree identification: In this scenario, sequences for one label are generated from a phylogenetic tree, whereas the other label identifies sequences that are either completely randomly generated or sampled from background sequence (and therefore without any guiding evolutionary relationship). This experiment tests whether the model is differentiating evolutionary processes from noise.
> >
> > We additionally tested two variations of synthetic datasets - one in which sequences were completely randomly generated, and one in which sequences are sampled from existing genomes to see if the fully artificial sequences we originally used were not generalizable to our real world dataset results. Overall, we constructed 5 additional experiments, the results of which we share in the new Table 2 in the paper and reproduce here (with full experimental details in A.6):
> >
> > | **Scenario** | **Sequence**  | **Euclidean CNN**      | **Hyperbolic HCNN-S**        | **Hyperbolic HCNN-M**        |
> > |--------------|---------------|-------------------------|------------------------------|------------------------------|
> > | **A**        | Artificial    | 62.38 ± 2.28           | **65.25 ± 3.27**             | 59.25 ± 2.60                |
> > |              | Real          | 61.72 ± 3.08           | **66.44 ± 3.14**             | 61.26 ± 2.99                |
> > | **B**        | Artificial    | 58.50 ± 0.82           | **60.53 ± 0.80**             | 59.75 ± 0.54                |
> > |              | Real          | 57.50 ± 0.88           | **62.53 ± 6.94**             | 59.12 ± 0.54                |
> > | **C**        | Artificial    | 62.05 ± 1.62           | **67.65 ± 1.09** †           | 67.43 ± 1.57 †              |
> > |              | Real          | 66.22 ± 0.44           | **73.62 ± 0.62** †           | 69.30 ± 2.34 †              |
> >
> >
> > We are pleased to report that the results of these experiments provide a clearer understanding of how the hyperbolic inductive bias functions in a learning context. In the genomics domain, **this bias primarily operates by uncovering the underlying phylogenetic tree structure amidst noise**. While this learning mechanism may also offer advantages in disentangling distinct evolutionary patterns (Scenario B), the results consistently indicate that excelling in Scenario C is somewhat orthogonal to excelling in Scenario A. This is likely because sequence differentiation is performed at the tree level rather than at the clade level. Notably, this finding aligns with results observed on real-world datasets: the most significant performance gains are observed in scenarios where an evolutionary signal (e.g., transcription factor binding sites, epigenetic marks, or transposable elements) is effectively distinguished from background noise (e.g., background sequences).
> >
> > Overall, we find these results to support our original hypothesis discussed in the introduction of our paper - the inductive biases of our models extend past approximating explicit edit distance mutations to instead focus on latent hierarchical structure that likely occupies a lower dimensional manifold.

---

> > > ### Author Response · Authors · 2024-11-19
> > > **Response to Reviewer H56n (part 3)**
> > >
> > > > W3: In genomic sequence modeling it is common to do homology splitting where sequences related through their homologous relationships are fully left out and the model's capacity is assessed on whether it's able to generalize to unseen homology branches. Do the authors expect that this inductive bias helps or hurts the performance of the model in this domain? Can they perform an experiment validating this?
> > >
> > > Thanks for raising a pertinent question. Our intuition after conducting the synthetic experiments in the prior comment is that our model’s inductive biases would improve generalizability to unseen homology branches. In order to validate this hypothesis, we assess the zero-shot capability of our model in identifying sequences originating from an unseen phylogenetic tree against random background sequences. For our training data, we generate a synthetic dataset as we did for testing Scenario C (sequences generated from the tree share the same label, and sequences not from the tree share a different label). However, instead of splitting this dataset into train/validation/test sequences, we create our test set separately by generating a completely new phylogenetic tree and sampling sequences from this set. The tree-generated sequences in the test dataset thus originate from completely unseen homology branches. This process is visualized in the new Figure 9 and further detailed in section A.7 of the paper.
> > >
> > > | CNN | HCNN-S | HCNN-M |
> > > | ------- |------- | ------- |
> > > | 24.31 &pm; 7.99 |  **45.73 &pm; 8.93** †  | 40.87 &pm; 8.93 † |
> > >
> > > † Significant with p < 0.05, Wilcoxon rank-sum test
> > >
> > > From our results, we observe that the hyperbolic models gain a significant advantage over the Euclidean model, which validates our hypothesis. This finding would suggest that the inductive biases of a hyperbolic model offer an even larger advantage over Euclidean models than originally estimated, since most genomic datasets do not account for this effect and may therefore overestimate performance of prediction methods.
> > >
> > > > Questions:
> > > > If the hierarchical relationship is between sequences then why do the authors note that a single point would be an individual nucleotide and a sequence would be a set? Isn't a sequence then exponential in the number of individual nucleotides thus resulting in an exponential growth of possibility space even without the hyperbolic space inductive bias?
> > >
> > > Thanks for raising this clarifying question - we realize we were not sufficiently clear in the paper. The hierarchical relationship is indeed between sequences, and thus the hyperboloid is learned **across** sequences, for any given nucleotide position. While we agree that the space of possible sequences is exponential in sequence length, we believe that hyperbolic spaces – in which neighborhoods grow exponentially rather than polynomially – better match this property of our data.
> > >
> > > > There is no mention regarding the parameter count of the eucledian CNN. I assume it is the same as the hyperbolic version but would be good to get a confirmation. Can the authors also include FLOP count per forward / backward pass for each of the models?
> > >
> > > We will add a version of the following table to our appendix, confirming the parameter count of the Euclidean CNN.
> > >
> > > | Model  | GFLOPs| relative FLOPs (compared to CNN)| Parameters |
> > > | -------- | ------- |------- | ------- |
> > > | HCNN-S | 3.7662 |0.97 | 6.6M |
> > > | HCNN-M  | 15.0649 | 3.87 | 6.6M |
> > > | CNN | 3.8966 | 1.00 | 6.6M |
> > >
> > > After computing the FLOP counts per forward/backward pass for each of the models, we see that the HCNN-S model performs slightly fewer FLOPs compared to the CNN, however the added mapping operations between manifolds in the HCNN-M increases its FLOPs. This may be a tradeoff to consider during model selection, and we will mention this in the paper.
> > >
> > > > The work can benefit from an expanded more thorough explanation in the appendix of mapping between tangent space and Lorentz manifold especially when describing the hypebolic layers.
> > >
> > > We have expanded section A.1 in the paper with a thorough explanation of mapping between the tangent space and the Lorentz manifold as they pertain to the hyperbolic layers (now section A.1.2).
> > >
> > > > Can the authors mention how the confidence intervals were generated in table 1.
> > >
> > > We compute the standard deviation across five runs, which we have clarified in the Table 1 caption.

---

> > > > ### Author Response · Authors · 2024-11-19
> > > > **Response to Reviewer H56n (part 4)**
> > > >
> > > > > Do the authors find it strange that for covid variant classification, arguably in a very clear cut case where there is abundant information regarding evolution of the virus the hyperbolic architecture doesn't perform so well? Do the authors have a hypothesis for why that is the case?
> > > >
> > > > Thanks for pointing this out - we also found the results from the covid variant classification task to be perplexing at first. However, upon further consideration, we posit two main reasons for why we observe this result:
> > > >
> > > > 1. The covid dataset is designed by sampling random portions of each covid variant genome. The underlying sequence-generating process behind covid variant evolution most closely resembles Scenario A in our synthetic dataset experiments - in which a single ancestral sequence evolves along multiple paths. Since the task is designed to differentiate sequences across covid variants, it is similar to differentiating separate clades on a tree. As we concluded from our earlier synthetic experiments in W2, the inductive biases of our model are better suited to identifying phylogenetic signal amidst noise than in differentiating within the phylogenetic tree.
> > > >
> > > > 2. Our investigation into the Gromov hyperbolicity distribution of the covid dataset shows that the delta estimates are notably the most right shifted across all the datasets, indicating that the data are not well suited for hyperbolic spaces. This incompatibility likely compounds the modeling results in point 1.
> > > >
> > > >
> > > > Again, thank you for your discerning questions and comments.

---

> > > > > ### Comment · Reviewer_H56n · 2024-11-22
> > > > >
> > > > > The authors perform a thorough rebuttal tying together in-silico mutagenesis experiments to performance on empirical tasks explaining lower performance on the Covid variant classification dataset. I particularly enjoyed figure 16 which provides an additional data point of the validity of the inductive bias. The addition of clarifying synthetic experiments (figure 2) significantly strengthens the argument. I have updated my score to reflect the improvements that the authors have made to the paper.

---

### Official Review · Reviewer_Q1G1 · 2024-11-03

**Soundness:** 2
**Presentation:** 2
**Contribution:** 2
**Rating:** 5
**Confidence:** 4

**Summary:**

This paper adopts the hyperbolic CNNs to genome representation learning. Empirical results suggest the advantage of hyperbolic CNN to its Euclidean equivalents in genome learning. Moreover, the hyperbolic CNNs for genome sequences can outperform existing genome foundation models on 7 out of 27 datasets on the GUE benchmark.

**Strengths:**

1. The paper is well-written, with good backgrounds and clear visualization.
2. The introduced Transposable Elements Benchmark is valuable to the research community.
3. The hyperbolic CNN demonstrates consistent improvements over the Euclidean CNN. It could serve as a baseline choice for genome classification models.

**Weaknesses:**

1. Comparing the number of parameters between CNN-based models and Transformer-based models is not convincing. Due to the underlying difference between these two model architectures, fewer parameters do not guarantee efficiency. A more suitable comparison would be the time and memory usage of training the models on the same dataset.
2. There is a lack of baselines. As a CNN-based model, the model should also be compared with HyenaDNA and Caduceus.
3. The empirical study is not well-presented. In the main text, this paper dedicates most of the experimental sections to the comparison of hyperbolic and Euclidean CNN, which is less relevant to the genomics audience. On the one hand, this conclusion has already been drawn in the HCNN paper [1]. On the other hand, the Euclidean CNN is not the SOTA in genome classification. Compared with existing genomics foundation models, as shown in Table 4, it is more important, in my opinion.
4. The contribution is limited from both methodology and application perspectives.
    - Methodologically, HCNN is introduced in ICLR 2024, and the advantage of hyperbolic over Euclidean equivalents is also shown in that paper. This work adopts the same model (with tiny modifications on two versions ) and validates the same observations in genome embedding.
    - From an application perspective, this work's contribution is limited. Though the authors do not report the wall-clock time / memory usage of the training the model, given that GFMs like NT and DNABERT only require 3-5 epochs to converge and this model is trained for 100 epochs, this model is likely to be more costly than existing ones. Moreover, it does not deliver better performances than GFMs. From an application perspective, users do not care much about whether a model is pre-trained / randomly initialized. The cost of fitting a model to a specific dataset is more meaningful.

[1] Bdeir, Ahmad, Kristian Schwethelm, and Niels Landwehr. "Fully Hyperbolic Convolutional Neural Networks for Computer Vision." arXiv preprint arXiv:2303.15919 (2023).

**Questions:**

1. Can you compare the cost of different models by the wall-clock time of fitting it on the same dataset with the same devices?

---

> ### Author Response · Authors · 2024-11-19
> **Response to Reviewer Q1G1 (part 1)**
>
> Thank you for taking the time to write this detailed review. We are glad to see that you enjoyed the writing and presentation of our work, as well as the background motivation. We also appreciate your recognition of the Transposable Elements Benchmark as a valuable contribution to the research community.
>
> Our initial paper was insufficiently clear in that **our main goal is to highlight the compatibility of the HCNN framework as specifically well suited to the genomics domain.**  While we ourselves do not present a GFM in this paper, we provide multiple lines of evidence that future GFMs/GMs may benefit from the inductive biases inherent in hyperbolic architectures. We believe that most of your concerns stem from comparisons made to existing GFMs, however our intention was to open up a new class of modeling approaches for genomics with our work. As such, if we have better recontextualized the core contribution of the paper and dispelled your concerns regarding model efficiency, we hope that you might be willing to raise your score. We hope that by starting a dialogue early, we will be able to address any responses you may have.
>
> We’ll now respond inline to specific comments:
> > Comparing the number of parameters between CNN-based models and Transformer-based models is not convincing. Due to the underlying difference between these two model architectures, fewer parameters do not guarantee efficiency. A more suitable comparison would be the time and memory usage of training the models on the same dataset.
>
> We completely agree that fewer parameters do not always guarantee efficiency. You raise an important point in terms of examining model efficiency across multiple criteria, which carries practical importance for the wider research community. In light of your comment, we compared the time, memory, and computational burden of training the four classes of models that achieved SOTA on GUE: NT-2500M multi, DNABERT-2, HCNN-S, and HCNN-M. Our results are summarized in the following table:
>
> | Model    | Memory (VRAM) | Time - Machine 1 | Time - Machine 2 | relative FLOPs|
> | -------- | ------- |------- |------- |------- |
> | NT-2500M-multi  | 192 GB (640 GB*) | N/A | 85 min 34 s (24.6) |7892 |
> | DNABERT-2 |  48 GB  | 9 min 7 s (2.25) | 9 min 30 sec (2.73) | 406 |
> | HCNN-M  | 8GB  | 4 min 23 s (1.08) | 3 min 41 s (1.06) | 3.87 |
> | HCNN-S | 8GB  | 4 min 3 s (1) | 3 min 29 s (1) |1.00 |
>
> Memory: We report the minimum required VRAM necessary to fit each model to the same GUE dataset. For HCNN-S and HCNN-M, we report the VRAM required to train the models from scratch, and for DNABERT-2 and NT-2500M-multi we report the VRAM necessary to finetune the model on the dataset. We use an asterisk to note that for NT-2500M-multi, the authors reported fine-tuning on eight A100 GPUs (640 GB), which is likely closer to the recommended VRAM [1].
>
> Time: We compute the wall-clock time by fitting our models on the same GUE dataset.  We compare times across two machines - a relatively smaller machine 1 (two RTX 3090s), and a larger machine 2 (four L40s) that can support NT-2500M-multi. We use the recommended 3 epochs for fine-tuning DNABERT-2/NT-2500M-multi per the DNABERT-2 repo, and use the default number of training epochs (100) for HCNN-S/HCNN-M. We additionally report the relative increase in time compared to the fastest model (HCNN-S) in parentheses next to the absolute time.
>
> FLOPs: We measure each model’s computational cost by considering the relative Floating Point Operations (FLOPs)—which is the total number of multiplication and addition operations during a forward and backward pass—compared to HCNN-S, which performs the lowest number of FLOPs.
>
> From the following experiments, we can conclude HCNNs maintain supremacy per all relevant criteria: parameters, runtime, computation cost, and memory. Across most criteria, the difference is often several orders of magnitude in improvement, cementing HCNNs as a viable resource for the genomics research community.
>
> > There is a lack of baselines. As a CNN-based model, the model should also be compared with HyenaDNA and Caduceus.
>
> Thanks for the suggestion - we agree that it may be helpful to include comparisons to diverse, non-Transformer/CNN architectures such as HyenaDNA and Caduceus. We have added HyenaDNA and Caduceus to Table 4 (which is now Table 5 in the revised paper). HCNNs outperform HyenaDNA on 27/28 datasets, and Caduceus on 19/28 datasets.

---

> > ### Author Response · Authors · 2024-11-19
> > **Response to Reviewer Q1G1 (part 2)**
> >
> > > The empirical study is not well-presented. In the main text, this paper dedicates most of the experimental sections to the comparison of hyperbolic and Euclidean CNN, which is less relevant to the genomics audience. On the one hand, this conclusion has already been drawn in the HCNN paper [1]. On the other hand, the Euclidean CNN is not the SOTA in genome classification. Compared with existing genomics foundation models, as shown in Table 4, it is more important, in my opinion.
> >
> > Thank you for raising these points. We have a number of clarifications in response to your concerns, and will integrate these ideas into the body of our paper where appropriate.
> >
> > 1. The primary motivation for our comparison between hyperbolic and Euclidean CNNs is to highlight the improvements made specifically from using a hyperbolic framework in the genomics domain which has not been tested comprehensively until now. We believe that this novelty will be of interest to the genomics audience.
> > 2. While we are beholden to the HCNN paper for enabling the modeling in our paper, the paper does not demonstrate strong advantages of HCNNs over Euclidean equivalents. Indeed, their primary contribution is their novel hyperbolic formulation, as their fully hyperbolic model offers tiny margins of improvement over the CNN model (accuracy increase of 0.35-0.52%) and in some cases does not even outperform the Euclidean model on image datasets  (Table 1 [5]).
> > 3. In fact, we believe that the Bdeir paper highlights how careful consideration is needed for the domains in which hyperbolic models are used, as they may not always confer an advantage. This largely guided our work in properly motivating the use of HCNNs in genomics.
> > 4. In building our HCNNs, we sought to establish reasonable baselines comparing different inductive biases, and did not optimize a model (with intensive ablations and hyperparameter search) to achieve SOTA on benchmark datasets as GFMs have. Therefore, we intended the results of Table 4 as a secondary finding that show our baselines are competitive with SOTA (highlighting the potential of HCNNs).
> > 5. Indeed, from Table 4 (now Table 5 in the updated manuscript), we observe that no single class of models reaches SOTA on the majority of GUE datasets. There are clearly tradeoffs between model classes that suit different tasks, and we highlight one set of promising approaches.
> > 6. While we agree with you that SOTA performance is an important consideration, we highlight the potential for HCNNs to be improve GMs that are CNN-based (such as GPN [2]) or CNN hybrids (such as Enformer [3]), which have already attained SOTA on genomics tasks. We believe this potential is of significant interest to the field.
> >
> > > The contribution is limited from both methodology and application perspectives.
> > > Methodologically, HCNN is introduced in ICLR 2024, and the advantage of hyperbolic over Euclidean equivalents is also shown in that paper. This work adopts the same model (with tiny modifications on two versions ) and validates the same observations in genome embedding.
> >
> > As stated in bullet #2 from the previous point, Bdeir et. al does not show a particularly compelling use case for HCNNs. However, we provide ample evidence specifically motivating the use of HCNNs in genomics, including an explanation of the evolutionary perspective, and a theoretical justification through our exploration of Gromov hyperbolicity. While we primarily consider our work to be an applications paper, we have also created a novel approach to evaluate the hyperbolicity of datasets, by considering the distributional shifts that occur in hyperbolicity both with respect to the underlying data generating processes and data dimensionality. We show that Gromov hyperbolicity correlates significantly with improvements in performance from a hyperbolic model, providing further evidence that hyperbolic models are well-suited to genomics.
> >
> > We hope you will also consider the six additional experiments we have introduced to further understand the biological processes captured by our models, detailed in the following sections of the paper:
> > 1. Section 4.1, Synthetic datasets, Figure 2, Table 2, Section A.6: we introduce and test multiple data-generating processes to determine how HCNNs capture biological signal.
> > 2. Section A.7, Figure 9, Table 6: We show that the inductive biases of HCNNs improve generalizability to unseen homology branches.
> >
> > These experiments make a novel contribution by providing a unified explanation for the specific advantages of hyperbolic models in genomics while also enhancing interpretability of the signal being learned.

---

> > > ### Author Response · Authors · 2024-11-19
> > > **Response to Reviewer Q1G1 (part 3)**
> > >
> > > > From an application perspective, this work's contribution is limited. Though the authors do not report the wall-clock time / memory usage of the training the model, given that GFMs like NT and DNABERT only require 3-5 epochs to converge and this model is trained for 100 epochs, this model is likely to be more costly than existing ones. Moreover, it does not deliver better performances than GFMs. From an application perspective, users do not care much about whether a model is pre-trained / randomly initialized. The cost of fitting a model to a specific dataset is more meaningful.
> > >
> > > We are grateful for your previous suggestions, which have enabled us to unequivocally demonstrate that our models are less costly than GFMs. Our experiments show that our models are generally **magnitudes** more efficient, and this calculus does not even consider the vast cost of pretraining, which can take weeks to months and potentially terabytes of RAM [1, 4], and which is frequently a necessary step when applying GFMs to new areas of genomics research (such as non-human species or individual genomes).
> > >
> > > To summarize, our work is a promising contribution because on one hand, **we present a lightweight, standalone class of models** that can be used to explore a diversity of genomics areas, and will thus **enable more researchers to iterate quickly on experiments/ideas**. On the other hand, **the architecture we present is highly modular and can be easily scaled** to create competitive GFMs.
> > >
> > >
> > > > Questions:
> > > > Can you compare the cost of different models by the wall-clock time of fitting it on the same dataset with the same devices?
> > >
> > > We provide the results of this comparison in our earlier response on model efficiency.
> > >
> > > [1] Dalla-Torre, H., Gonzalez, L., Mendoza-Revilla, J., Carranza, N. L., Grzywaczewski, A. H., Oteri, F., ... & Pierrot, T. (2023). The nucleotide transformer: Building and evaluating robust foundation models for human genomics. BioRxiv, 2023-01.
> > >
> > > [2] Benegas, G., Batra, S. S., & Song, Y. S. (2023). DNA language models are powerful predictors of genome-wide variant effects. Proceedings of the National Academy of Sciences, 120(44), e2311219120.
> > >
> > > [3] Avsec, Ž., Agarwal, V., Visentin, D., Ledsam, J. R., Grabska-Barwinska, A., Taylor, K. R., ... & Kelley, D. R. (2021). Effective gene expression prediction from sequence by integrating long-range interactions. Nature methods, 18(10), 1196-1203.
> > >
> > > [4] Zhihan Zhou, Yanrong Ji, Weijian Li, Pratik Dutta, Ramana V. Davuluri, and Han Liu. DNABERT-2: efficient foundation model and benchmark for multi-species genomes. In The Twelfth International Conference on Learning Representations, ICLR 2024, Vienna, Austria, May 7-11, 2024.
> > >
> > > [5] Bdeir, Ahmad, Kristian Schwethelm, and Niels Landwehr. Fully Hyperbolic Convolutional Neural Networks for Computer Vision. In The Twelfth International Conference on Learning Representations, ICLR 2024, Vienna, Austria, May 7-11, 2024.

---

> > > > ### Comment · Reviewer_Q1G1 · 2024-11-22
> > > > **Raise my score to 5.**
> > > >
> > > > Thank you for your detailed and insightful responses. Your responses solve most of my previous concerns. I agree that your work has demonstrated the potential of the hyperbolic framework in the genomics domain and that your experiments with time/memory analysis are convincing. I think this work could be largely enhanced if the authors can train a genome foundation model with the hyperbolic framework and demonstrate its capability. For the current version, I have increased my score from 3 to 5 to reflect my latest judgment. I am okay with accepting this paper if other reviewers recommend it.

---

### Official Review · Reviewer_Zknz · 2024-11-03

**Soundness:** 3
**Presentation:** 2
**Contribution:** 2
**Rating:** 5
**Confidence:** 3

**Summary:**

The authors introduce a hyperbolic CNN for genomic sequences. They compare their model to a standard CNN on both synthetic and real benchmarks for genomic sequence classification (GUE, GE, and their transposable element benchmark). The authors find that the HCNN model outperforms the standard CNN on most tasks.

**Strengths:**

Hyperbolic architectures is a promising avenue for training models on genomic sequences, as they are inherently related through phylogenetic tree structures. This paper provides interesting initial results in this direction.

**Weaknesses:**

- How does the CNN/HCNN performance scale with number of parameters? Figure 5 seems to indicate that performance gets worse with increasing hidden dim? Additionally figure 5 should include error bars. Also the cumulative improvement y-axis is unclear. It should just be the average MCC value.

- Figure 1 shows that sequences which obey a phylogenetic tree structure are embedded into a hyperbolic space which learns this structure. However I do not see any results which indicate that the phylogenetic structure is being learned. The authors could provide an experiment to validate this by looking at correlation between embedding distances and phylogenetic distances (for example EDS task from https://www.biorxiv.org/content/10.1101/2024.07.10.602933v1)

- The results would be stronger if the authors provided a baseline column for a standard transformer architecture in Table 1, as CNNs are used less frequently for genomic tasks. Additionally, the DNABERT/NT performance should be included in Table 1, not in supplementary Table 4, and would be much more clear by averaging a single value for each task.

- Figure 2 shows the decision boundary for a 2D model, and shows better separation for the hyperbolic embeddings. However, it is not clear that this result extends to higher dimensions. Instead, the figure would be stronger by using a PCA/UMAP plot with a higher dimensionality model.

**Questions:**

- Is the CNN/HCNN model pretrained with a MLM task similar to DNABERT models? If so, on what dataset? If not, why?

---

> ### Author Response · Authors · 2024-11-24
> **Response to Reviewer Zknz (part 1)**
>
> Thank you for taking the time to review our work. We appreciate your recognition of hyperbolic architectures as a promising avenue for genome sequence modeling.
>
> > How does the CNN/HCNN performance scale with number of parameters? Figure 5 seems to indicate that performance gets worse with increasing hidden dim?
>
> Thanks for raising this point - we would like to clarify that Figure 5 shows the average net increase in performance is higher at lower dimensions when using HCNNs **compared to** CNNs.  Given the importance of learning compact representations in all deep learning domains, we intended this figure to demonstrate that HCNNs offer an added benefit in this area.  We have added a second plot to Figure 5 that shows that HCNN/CNN performance improves with increasing hidden dimension size.
>
> > Additionally figure 5 should include error bars. Also the cumulative improvement y-axis is unclear. It should just be the average MCC value.
>
> We agree that this change would add clarity to the figure, so we have updated it to show the average MCC value with error bars.
>
> > Figure 1 shows that sequences which obey a phylogenetic tree structure are embedded into a hyperbolic space which learns this structure. However I do not see any results which indicate that the phylogenetic structure is being learned.
>
> Thank you for raising an intriguing and relevant point. We have conducted a number of additional experiments to elucidate the ways in which our models learn phylogenetic structure (experiment 1), provide interpretability for the learned phylogenetic embeddings (experiment 2), and demonstrate a concrete example of how our learning process confers advantages for phylogenetic data over existing models (experiment 3). We summarize the contributions by experiment here:
>
> **Exp 1).** Creating synthetic datasets using an evolutionary process simulator (section 4.1): we extend our synthetic dataset experiment to investigate three potential scenarios underlying the data-generating processes of sequences. We determine that our models primarily operate by uncovering underlying phylogenetic tree structure amidst noise.
>
> **Exp 2).** In silico mutagenesis (section A.10): We investigate how the phylogenetic structure of sequences are represented by HCNNs through a perturbation experiment. We find that sequences near the top of the learned hierarchical structure represent ambiguous sequences, and ablating conserved regions of sequences results in the loss of definitive features, shifting them closer to the top of the hierarchy.
>
> **Exp 3).** Homology splitting (section A.7): We show that the inductive biases of HCNNs enable improved generalizability to unseen homology branches, which is an important learning priority for all biological sequence models.
>
> We further elaborate on each experiment to provide necessary details. For experiment 1, we reconsidered the various plausible data-generating processes for biological sequences. We defined three potential cases of biological signal transmission that might be learned by the models, and constructed 3 classes of synthetic datasets (visualized in the new Figure 2):
>
> (A). Intra-tree differentiation: this was our original synthetic experiment. In this scenario, we generated sequences from a single phylogenetic tree, with labels derived from clade membership. This experiment tested whether a hyperbolic model would better differentiate parts of a single phylogenetic tree from each other.
>
> (B) Inter-tree differentiation: in this scenario sequences are generated from different phylogenetic trees, with labels derived from phylogeny membership. This experiment tests whether the model is differentiating different evolutionary processes from one another.
>
> (C) Tree identification: In this scenario, sequences for one label are generated from a phylogenetic tree, whereas the other label identifies sequences that are either completely randomly generated or sampled from background sequence (and therefore without any guiding evolutionary relationship). This experiment tests whether the model is differentiating evolutionary processes from noise.
>
> We additionally tested two variations of synthetic datasets - one in which sequences were completely randomly generated, and one in which sequences are sampled from existing genomes to see if the fully artificial sequences we originally used were not generalizable to our real world dataset results.

---

> ### Author Response · Authors · 2024-11-24
> **Response to Reviewer Zknz (part 2)**
>
> Results are reported here:
>
> | **Scenario** | **Sequence**  | **Euclidean CNN**      | **Hyperbolic HCNN-S**        | **Hyperbolic HCNN-M**        |
> |--------------|---------------|-------------------------|------------------------------|------------------------------|
> | **A**        | Artificial    | 62.38 ± 2.28           | **65.25 ± 3.27**             | 59.25 ± 2.60                |
> |              | Real          | 61.72 ± 3.08           | **66.44 ± 3.14**             | 61.26 ± 2.99                |
> | **B**        | Artificial    | 58.50 ± 0.82           | **60.53 ± 0.80**             | 59.75 ± 0.54                |
> |              | Real          | 57.50 ± 0.88           | **62.53 ± 6.94**             | 59.12 ± 0.54                |
> | **C**        | Artificial    | 62.05 ± 1.62           | **67.65 ± 1.09** †           | 67.43 ± 1.57 †              |
> |              | Real          | 66.22 ± 0.44           | **73.62 ± 0.62** †           | 69.30 ± 2.34 †              |
>
>
> We are pleased to report that the results of these experiments provide a clearer understanding of how phylogenetic tree structure is learned. Our models primarily operate by uncovering the underlying phylogenetic tree structure amidst noise. While this learning mechanism may also offer advantages in disentangling distinct evolutionary patterns (Scenario B), the results consistently indicate that excelling in Scenario C is somewhat orthogonal to excelling in Scenario A. This is likely because sequence differentiation is performed at the tree level rather than at the clade level. Notably, this finding aligns with results observed on real-world datasets: the most significant performance gains are observed in scenarios where an evolutionary signal (e.g., transcription factor binding sites, epigenetic marks, or transposable elements) is effectively distinguished from background noise (e.g., background sequences).
>
> Overall, we find these results to support our original hypothesis discussed in the introduction of our paper - the inductive biases of our models extend past approximating explicit edit distance mutations to instead focus on latent hierarchical structure that likely occupies a lower dimensional manifold.
>
> For experiment 2, we observe that in the processed pseudogene dataset in TEB, the sequence embeddings that lie close to the center of the Poincaré disk (the top of the hierarchy) correspond to low confidence embeddings for HCNNs (Figure 15), while the embeddings near the disk boundaries show the highest classification confidence. This is consistent with the idea that well defined sequences are at the bottom of the hierarchy where there is more space to separate out nuanced differences between sequences based on distinctive features.  Since we cannot readily identify features from DNA sequence, we conduct an experiment to dissect the sequence features informing the hyperbolic genome embedding. Given our dataset of processed pseudogenes, we examine the changes made to the HCNN representation by perturbing a fixed pseudogene sequence. For a fixed sequence, we:
>
> 1. Compute the Genomic Evolutionary Rate Profiling (GERP) [1] score for each nucleotide along the sequence. GERP scores quantify evolutionary constraints at specific genomic positions, identifying which positions are functionally important based on selective pressure.
> 2. Mutate a fraction of the nucleotides under the highest selective pressure
> 3. Use our HCNN to generate an embedding for this perturbed instance of our original sequence.
>
> Figure 16 visualizes this experiment using a processed pseudogene sequence and a background sequence. As the evolutionary signal under strong selection is eroded by the introduced mutations, it is likely that the features that make the pseudogene more “gene-like” are degraded. This degradation ultimately makes the sequence more ambiguous to the HCNN, and we observe that the perturbed representations move closer to the top of the hierarchy (near the center of the Poincaré disk), where the low confidence sequences lie. Removal of these evolutionary features actively hinders the model in identifying pseudogenes. However, perturbing conserved regions from the noisy background sequences does not appear to have this effect, as the model focuses on learning features common to the pseudogene class.
>
> [1] Cooper, G. M., Stone, E. A., Asimenos, G., Green, E. D., Batzoglou, S., & Sidow, A. (2005). Distribution and intensity of constraint in mammalian genomic sequence. Genome research, 15(7), 901-913.

---

> > ### Author Response · Authors · 2024-11-24
> > **Response to Reviewer Zknz (part 3)**
> >
> > For experiment 3, we performed an experiment to determine whether HCNNs improve generalizability to unseen homology branches. We assessed the zero-shot capability of our model in identifying sequences originating from an unseen phylogenetic tree against random background sequences. For our training data, we generate a synthetic dataset as we did for testing Scenario C (sequences generated from the tree share the same label, and sequences not from the tree share a different label). However, instead of splitting this dataset into train/validation/test sequences, we create our test set separately by generating a completely new phylogenetic tree and sampling sequences from this set. This process is visualized in the new Figure 9.
> >
> > | CNN | HCNN-S | HCNN-M |
> > | ------- |------- | ------- |
> > | 24.31 &pm; 7.99 |  **45.73 &pm; 8.93** † | 40.87 &pm; 8.93 † |
> >
> > †Significant with p < 0.05, Wilcoxon rank-sum test
> >
> > From our results, we observe that the hyperbolic models gain a significant advantage over the Euclidean model. This finding would suggest that the inductive biases of a hyperbolic model offer an even larger advantage over Euclidean models than originally estimated, since most genomic datasets do not account for this effect and may therefore overestimate performance of prediction methods.
> >
> > Overall, we find that these three experiments have greatly unified our understanding of how phylogenetic tree structure is learned by our models.
> >
> > > The authors could provide an experiment to validate this by looking at correlation between embedding distances and phylogenetic distances (for example EDS task from https://www.biorxiv.org/content/10.1101/2024.07.10.602933v1)
> >
> > We appreciate that you offered a potential starting point for our experiments with the EDS task from [1]. Unfortunately, we could not use these datasets since they assume pretrained models and therefore only include a test set (no training/validation).
> >
> > [1] West-Roberts, J., Kravitz, J., Jha, N., Cornman, A., & Hwang, Y. (2024). Diverse Genomic Embedding Benchmark for functional evaluation across the tree of life. bioRxiv, 2024-07.
> >
> > > The results would be stronger if the authors provided a baseline column for a standard transformer architecture in Table 1, as CNNs are used less frequently for genomic tasks.
> > Additionally, the DNABERT/NT performance should be included in Table 1, not in supplementary Table 4, and would be much more clear by averaging a single value for each task.
> >
> > We would like to clarify that since our main aim is in showcasing the benefit of using hyperbolic geometry over Euclidean geometry-based operations, we compared HCNNs to CNNs to control for any geometry-agnostic differences in model architecture. We sought to establish reasonable baselines comparing different inductive biases, and did not optimize a model (with intensive ablations and hyperparameter search) to achieve SOTA on benchmark datasets as genome foundation models have. Therefore, we intended the results of Table 4 as a secondary finding that show our baselines are competitive with SOTA (highlighting the potential of HCNNs). Still, we agree that averaging values by task improves the clarity of Table 1, thus we have created Table 9 in the paper, which presents the performances averaged at the task level (and will work on integrating into the main text for the camera-ready).
> >
> > > Figure 2 shows the decision boundary for a 2D model, and shows better separation for the hyperbolic embeddings. However, it is not clear that this result extends to higher dimensions. Instead, the figure would be stronger by using a PCA/UMAP plot with a higher dimensionality model.
> >
> > Thanks for the suggestion. We have included UMAP plots using the higher dimensionality models we use in our benchmarks as Figure 17. We use the default UMAP output metric (“euclidean”) for the Euclidean CNN, and the hyperboloid output metric for the HCNN (plotting these results on the Poincaré disk). From these plots, we observe that embeddings from high dimensional HCNNs still appear to have greater separability between sequence classes when compared to the CNN embeddings. Notably, we see that the hyperbolic embeddings take advantage of the full embedding space and are positioned near the boundaries of the Poincaré disk for maximum separability.
> >
> > > Is the CNN/HCNN model pretrained with a MLM task similar to DNABERT models? If so, on what dataset? If not, why?
> >
> > We did not pretrain our models, instead training each model end-to-end on each dataset. As the main focus of this work is to demonstrate the specific advantages of using HCNNs in genomics, we did not find pretraining a necessary step in our approach. However, we believe that you raise an interesting point, and pretraining will likely improve the overall performance of the CNN/HCNNs. We leave this as a direction for future research, and have rewritten the conclusion to include this idea.

---

> > > ### Comment · Reviewer_Zknz · 2024-11-25
> > >
> > > Thank you for the thorough response and additional experiments. The authors have resolved my main questions from the review. I agree with reviewer Q1G1 that the work would be improved if the authors could pretrain a model using the hyperbolic framework and demonstrate its capability compared to existing models such as ESM2 (even at the 8M or 35M parameter scale), and evaluate particularly for zero-shot prediction of phylogenetic distances.
> > >
> > > I understand that this is likely beyond the scope of the current work, which focuses on motivating hyperbolic embeddings for biological sequences, primarily on synthetic datasets. As it stands, this work presents many exploratory experiments and analyses to motivate future work on hyperbolic embeddings, which I believe is important, but does not yet demonstrate the usefulness of these models compared to existing euclidian models like ESM, ProGen, NucleotideTransformer, etc.

---

### Author Response · Authors · 2024-12-04

We thank the reviewers for an engaging and productive discussion. While we’ve addressed all of the reviewers’ main concerns, reviewers Q1G1 and Zknz raised an additional point about pretraining a genome foundation model (GFM) to compare its performance against existing GFMs.  Although a pretrained GFM presents an interesting line of research as a follow up to this paper, our findings comprise a necessary standalone foundational work for the development of hyperbolic GFMs. Still, recognizing the importance of scaling for future GFMs, we used the rebuttal period to scale HCNN-S and HCNN-M to a parameter count more consistent with GFMs (increasing channel dimension to 256 and kernel size to 15, without hyperparameter tuning due to time constraints). While these larger HCNNs are still smaller than their competitors, they reach SOTA performance on more GUE datasets than DNABERT-2 and NT-2500M-multi, the two GFMs that attain SOTA performance on GUE. These findings, presented below, demonstrate the capabilities of the hyperbolic framework.

| Model Class           |SOTA performance on GUE (number of datasets) |
|-------------------|--------------------:|
| NT-2500M-multi |  5 |
| DNABERT-2 | 11 |
| HCNNs (Large) |     **12** |


Full GUE results are shown here:

|          | NT-2500M-multi | DNABERT-2 | DNABERT-2-PT | HCNN-S (Large) | HCNN-M (Large) |
|-------------------|----------------:|----------:|-------------:|---------------:|---------------:|
| Parameters            |           2.5B |       117M |         117M |            **86M** |            **86M** |
| H3                |          78.77 |      78.27 |        **80.17** |          72.17 |          72.21 |
| H3K14ac           |          56.20 |      52.57 |        57.42 |          **70.56** |          69.87 |
| H3K36me3          |          61.99 |      56.88 |        61.90 |          **68.06** |          67.92 |
| H3K4me1           |          55.30 |      50.52 |        53.00 |          53.33 |          **55.45** |
| H3K4me2           |          36.49 |      31.13 |        39.89 |          **54.67** |          52.50 |
| H3K4me3           |          40.34 |      36.27 |        41.20 |          **67.25** |          64.61 |
| H3K79me3          |          64.70 |      67.39 |        65.46 |          70.49 |          **70.65** |
| H3K9ac            |          56.01 |      55.63 |        57.07 |          **63.36** |          60.66 |
| H4                |          81.67 |      80.71 |        **81.86** |          76.54 |          74.78 |
| H4ac              |          49.13 |      50.43 |        50.35 |          62.16 |          **67.30** |
| promoter all               |          **91.01** |      86.77 |        88.31 |          83.35 |          83.73 |
| promoter notata            |          94.00 |      94.27 |        **94.34** |          88.36 |          90.67 |
| promoter tata              |          79.43 |      71.59 |        68.79 |          **81.86** |          79.19 |
| Human TF 0                 |          66.64 |      **71.99** |        69.12 |          65.09 |          62.85 |
| Human TF 1                 |          70.28 |      **76.06** |        71.87 |          67.59 |          69.91 |
| Human TF 2                 |          58.72 |      66.52 |        62.96 |          **70.73** |          63.79 |
| Human TF 3                 |          51.65 |      **58.54** |        55.35 |          42.12 |          46.26 |
| Human TF 4                 |          69.34 |      **77.43** |        74.94 |          71.95 |          70.33 |
| core promoter all          |          **70.33** |      69.37 |        67.50 |          61.77 |          62.56 |
| core promoter notata       |          **71.58** |      68.04 |        69.53 |          66.01 |          65.71 |
| core promoter tata         |          72.97 |      74.17 |        76.18 |          80.20 |          **80.26** |
| Mouse TF 0        |          63.31 |      56.76 |        **64.23** |          48.84 |          49.24 |
| Mouse TF 1        |          83.76 |      84.77 |        **86.28** |          81.07 |          79.43 |
| Mouse TF 2        |          71.52 |      79.32 |        **81.28** |          80.51 |          75.61 |
| Mouse TF 3        |          69.44 |      66.47 |        73.49 |          **81.57** |          78.77 |
| Mouse TF 4        |          47.07 |      **52.66** |        50.80 |          41.79 |          43.70 |
| Covid             |          **73.04** |      71.02 |        68.49 |          45.06 |          31.09 |
| Splice            |          **89.35** |      84.99 |        85.93 |          79.23 |          78.84 |

---

### Meta-Review · Area_Chair_YkK9 · 2024-12-21

**Metareview:**

This paper proposes an interesting method to represent genomic sequences. It is well-motivated, as it takes into account not only the sequence itself but the broader context of its phylogenetic relations -- a type of structure more easily captured by hyperbolic geometry.

As it stands, there was no consensus to be had: two Reviewers were suggesting acceptance and two Reviewers suggesting rejection. This is a borderline situation, in which I needed to make a judgement call.

From what I was able to see, the key remaining quip of the rejecting Reviewers (Q1G1 and Zknz) appears to be that the Authors did not attempt any inter-operability study of their method with foundation-level genomic models.

While I agree this would be a useful experiment to have in the arsenal, after further championing from the accepting Reviewers, it is evident that HCNN is performant on relevant genomic tasks without any integration into a pre-trained method. This speaks clear volumes about the practical applicability of the work and, especially considering it is an applied paper, and the associated costs of training foundation models, perhaps this is reason enough to let slide that more thorough foundational model integrations had not been attempted.

A such, I believe the paper is in a good shape to be presented at ICLR, and will recommend acceptance. Congratulations to the Authors!

**Additional Comments On Reviewer Discussion:**

The key part in making my final decision was the set of scaled-up results shared in the latest general comment posted by the Authors (on 4 December). Including them will certainly strengthen the paper! The rejecting Reviewers did not explicitly acknowledge or respond to these updates, so I made a judgement call that they would satisfy them.

---

### Decision · Program_Chairs · 2025-01-22

Accept (Poster)